# Stable Gradients for Stable Learning at Scale in Deep Reinforcement Learning

Roger Creus Castanyer[1,2]*  Johan Obando-Ceron[1,2]*  Lu Li[1,2]  Pierre-Luc Bacon[1,2]
Glen Berseth[1,2]  Aaron Courville[1,2]  Pablo Samuel Castro[1,2,3]

[1] Mila - Québec AI Institute  [2] Université de Montréal  [3] Google DeepMind

## Abstract

Scaling deep reinforcement learning networks is challenging and often results in degraded performance, yet the root causes of this failure mode remain poorly understood. Several recent works have proposed mechanisms to address this, but they are often complex and fail to highlight the causes underlying this difficulty. In this work, we conduct a series of empirical analyses which suggest that the combination of non-stationarity with gradient pathologies, due to suboptimal architectural choices, underlie the challenges of scale. We propose a series of direct interventions that stabilize gradient flow, enabling robust performance across a range of network depths and widths. Our interventions are simple to implement and compatible with well-established algorithms, and result in an effective mechanism that enables strong performance even at large scales. We validate our findings on a variety of agents and suites of environments. **Source code here.**

*"We must be able to look at the world and see it as a dynamic process, not a static picture."*
*— David Bohm*

## 1 Introduction

Recent advances in deep reinforcement learning (deep RL) have demonstrated the ability of deep neural networks to solve complex decision-making tasks from robotics to game play and resource optimization [Mnih et al., 2015, Vinyals et al., 2019, Bellemare et al., 2020, Fawzi et al., 2022]. Motivated by successes in supervised and generative learning, recent works have explored scaling architectures in deep RL, showing gains in representation quality and generalization across tasks [Farebrother et al., 2023, Taiga et al., 2023]. However, scaling neural networks in deep RL remains fundamentally challenging [Ceron et al., 2024b,a]. A central cause of this instability lies in the unique optimization challenges of RL. Unlike supervised learning, where data distributions are fixed, deep RL involves policy-dependent data that constantly change during training [Lyle et al., 2022]. Each update of the policy $\pi_\theta$ alters future states and rewards, making the training objective inherently *non-stationary*. Value-based methods exacerbate these issues via bootstrapping, recursively using predicted values as targets.

Estimation errors compound over time [Fujimoto et al., 2018], especially under sparse or delayed rewards [Zheng et al., 2018], leading to unstable updates, policy collapse, or value divergence [Van Hasselt et al., 2016, Lyle et al., 2023, 2024]. These dynamics are tightly coupled with architectural vulnerabilities. Deep networks face well known pathologies such as vanishing/exploding gradients [Pascanu et al., 2013], ill-conditioned Jacobians [Pennington et al., 2017], and activation saturation [Glorot and Bengio, 2010]. In deep RL, these are magnified by the "*deadly triad*" [Sutton and Barto, 2018, Van Hasselt et al., 2018], off-policy corrections, and changing targets. As networks

---

*Equal contribution. Correspondence to: Roger Creus C <roger.creus-castanyer@mila.quebec>,
Johan Obando-Ceron < jobando0730@gmail.com>, Pablo Samuel Castro <psc@google.com>

39th Conference on Neural Information Processing Systems (NeurIPS 2025).

scale, the risk of signal distortion and misalignment increases, resulting in underutilized capacity and brittle learning [Obando Ceron et al., 2023, Ceron et al., 2024a].

One overlooked source of these failures lies in how gradients propagate through the network. Specifically, the gradient decomposition, the layer-wise structure of backpropagation as a chain of Jacobians and weights determine how information flows during learning [Lee et al., 2020]. While gradient signal preservation has been studied in supervised learning [Schoenholz et al., 2017, Jacot et al., 2018], its role in deep RL, where both inputs and targets shift continually, remains poorly understood.

In this work, we investigate how gradient decomposition interacts with non-stationarity and network scaling in deep RL. We demonstrate that in non-stationary settings like RL – where targets are bootstrapped, policies evolve continually, and data distributions shift – gradient signals progressively degrade across depth. This motivates the need for methods that explicitly preserve the structure of gradient information across layers. We explore this through a series of controlled experiments and ablations across multiple algorithms and environments, demonstrating that actively encouraging gradient propagation significantly improves stability and performance, even with large networks. Our work offers a promising approach for scaling deep RL architectures, yielding substantial performance gains across a variety of agents and training regimes.

## 2   Preliminaries

**Deep Reinforcement Learning**   A deep reinforcement learning agent interacts with an environment through sequences of actions ($a \in \mathcal{A}$), which produce corresponding sequences of observations ($s \in \mathcal{S}$) and rewards ($r \in \mathbb{R}$), resulting in trajectories of the form $\tau := \{s_0, a_0, r_0, s_1, a_1, r_2, \ldots\}$. The agent's behavior is often represented by a neural network with parameters $\theta$, composed of convolutional layers $\{\phi_1, \phi_2, \ldots, \phi_{L_c}\}$ and dense (fully connected) layers $\{\psi_1, \psi_2, \ldots, \psi_{L_d}\}$, where $\psi_{L_d}$ has an output dimensionality of $|\mathcal{A}|$. At every timestep $t$, an observation $s_t \in \mathcal{S}$ is fed through the network to obtain an estimate of the long-term value of each action: $Q_\theta(s_t, \cdot) = \psi_{L_d}(\psi_{L_d-1}(\ldots(\phi_{L_c}(\ldots(\phi_1(s_t))\ldots))\ldots))$. The agent's policy $\pi_\theta(\cdot \mid s_t)$ specifies the probability of selecting each action, for instance by taking the softmax over the estimated values as in Eq. 1. The training objective is typically defined as the maximization of expected cumulative reward as in Eq. 2,

$$\pi_\theta(a_t \mid s_t) = \frac{e^{Q_\theta(s_t, a_t)}}{\sum_{a \in \mathcal{A}} e^{Q_\theta(s_t, a)}} \qquad (1) \qquad\qquad J(\theta) = \mathbb{E}_{\tau \sim \pi_\theta}\left[\sum_{t=0}^{\infty} \gamma^t r_t\right] \qquad (2)$$

where $\gamma \in [0, 1)$ is a discount factor and $\tau$ denotes a trajectory generated by following policy $\pi_\theta$. Optimization proceeds by minimizing a surrogate loss $\mathcal{L}(\theta)$, which may be derived from temporal-difference (TD) errors, policy gradients, or actor-critic estimators [Sutton and Barto, 2018]. In TD-based methods, the TD error at timestep $t$ is defined as:

$$\delta_t = r_t + \gamma V_\theta(s_{t+1}) - V_\theta(s_t),$$

where $V_\theta(s) = \mathbb{E}_{a \sim \pi_\theta(a|s)} Q_\theta(s, a)$. The recurrent nature of $\delta_t$ introduces dependencies on both current estimates and future rewards, making $\mathcal{L}(\theta)$ inherently non-stationary. As the policy $\pi_\theta$ evolves, the data distribution used for training shifts, further complicating optimization. Training is performed by collecting trajectories, computing gradients $\nabla \mathcal{L}(\theta)$, and updating parameters via $\theta \leftarrow \theta - \eta \nabla \mathcal{L}(\theta)$, where $\eta > 0$ is the learning rate. Following conventions from supervised learning, deep RL algorithms often use adaptive variants of stochastic gradient descent, such as Adam [Kingma and Ba, 2014] or RMSprop [Hinton, 2012], which adjust learning rates based on running estimates of gradient statistics. The gradients with respect to each layer are denoted by;

$$\nabla \phi_i = \frac{\partial \mathcal{L}}{\partial \phi_i}, \quad \nabla \psi_j = \frac{\partial \mathcal{L}}{\partial \psi_j},$$

where $\phi_i$ and $\psi_j$ represent the parameters (i.e., weight matrices or bias vectors) of layer $i$ and $j$ respectively. The structure and magnitude of these gradients ($\nabla \phi_i$ and $\nabla \psi_j$) are influenced by the loss function, data distribution collected from the environment, and the architecture itself. These per-layer gradients determine how effectively different parts of the network adapt during training.

While training large models in supervised learning settings present challenges, advances in initialization, normalization, and scaling strategies have enabled relatively stable optimization [Ioffe and

Szegedy, 2015, Ba et al., 2016, Glorot and Bengio, 2010]. Scaling up model size has been a central driver of progress across domains, improving generalization, enhancing representation learning, and boosting downstream performance [Kaplan et al., 2020].

Deep RL differs substantially from supervised learning. First, the data distribution is non-stationary, continually shifting as $\pi_\theta$ updates. Second, learning signals are often sparse, delayed, or noisy, which introduces variance in the estimated gradients [Han et al., 2022, Fujimoto et al., 2018]. These factors destabilize optimization and lead to loss surfaces with sharp curvature and complex local structure [Ilyas et al., 2020, Achiam et al., 2019]. Moreover, increasing model capacity often degrades performance unless regularization or architectural interventions are applied [Gogianu et al., 2021, Bjorck et al., 2021, Schwarzer et al., 2023, Wang et al., 2025].

These challenges are further compounded by both architectural and environmental factors. Network depth, width, initialization, and nonlinearity affect how gradients are propagated across layers. Meanwhile, reward sparsity, exploration dificulty, and transition stochasticity impose additional structure on the optimization landscape. The resulting geometry reflects the joint dynamics of policy, environment, and architecture, making deep RL optimization uniquely complex.

**Gradient Propagation**    Training deep networks poses fundamental challenges for effective gradient propagation [Glorot and Bengio, 2010]. As network depth increases, gradients may either vanish or explode as they are backpropagated through multiple layers, impeding the optimization of early layers and destabilizing learning dynamics [Ba et al., 2016]. These issues arise from repeated applications of the chain rule. For a network with intermediate hidden representations $\{h_0, h_1, \ldots, h_L\}$, where $h_k \in \mathbb{R}^{d_k}$, the gradient of the loss $\mathcal{L}$ with respect to a hidden layer $h_\ell$ is:

$$\frac{\partial \mathcal{L}}{\partial h_\ell} = \left( \prod_{k=\ell+1}^{L} \frac{\partial h_k}{\partial h_{k-1}} \right) \frac{\partial \mathcal{L}}{\partial h_L},$$

where each $\frac{\partial h_k}{\partial h_{k-1}} \in \mathbb{R}^{d_k \times d_{k-1}}$ is the Jacobian. If the singular values of these Jacobians are not properly controlled, their repeated multiplication can cause the norm of the gradient to shrink or grow exponentially with $\mathcal{L}$. This severely impairs convergence, as earlier layers receive little to no useful gradient signal or become numerically unstable [Ioffe and Szegedy, 2015, He et al., 2016].

In addition to depth, the width of the network also influences gradient propagation. Consider a fully connected layer with weight matrix $W \in \mathbb{R}^{m \times n}$ and input vector $h \in \mathbb{R}^n$. The output is $Wh \in \mathbb{R}^m$, and under the assumption that $W$ and $h$ have i.i.d. zero-mean entries with finite variance $\sigma_W$ and $\sigma_h$, respectively, the variance of the output is given by $\text{Var}[Wh] = n\,\sigma_W\,\sigma_h$. Thus, scaling the width $n$ without adjusting $\sigma_W$ and $\sigma_h$ leads to instability in forward and backward signal propagation affecting gradient norms and optimization trajectories.

Beyond depth and width, the choice of nonlinearity also plays a central role in determining how gradients propagate . In a typical feedforward network, hidden activations evolve as $h_k = \zeta(W_k h_{k-1})$, where $\zeta(\cdot)$ is a nonlinear activation function (e.g., ReLU, $\tanh$, sigmoid), and $W_k$ is the weight matrix at layer $k$. During backpropagation, the gradient with respect to a hidden layer includes the product of the Jacobian of the linear transformation and the derivative of the nonlinearity:

$$\frac{\partial \mathcal{L}}{\partial h_{k-1}} = W_k^\top \left( \zeta'(W_k h_{k-1}) \odot \frac{\partial \mathcal{L}}{\partial h_k} \right),$$

where $\zeta'(\cdot)$ denotes the elementwise derivative of the activation function, and $\odot$ represents elementwise multiplication. For ReLU, $\zeta'(x) = \mathbf{1}_{x>0}$, so the gradient is entirely blocked wherever the neuron is inactive. This leads to the well-known *dying ReLU* problem, where a significant portion of the network ceases to update and becomes untrainable [Lu et al., 2019, Shin and Karniadakis, 2020].

## 3   Diagnosis: Gradients Under Non-stationarity and Scale

A fundamental premise of modern deep learning is that scaling model capacity yields consistent gains in performance [Kaplan et al., 2020, Chowdhery et al., 2023]. This has held true in large-scale supervised learning, where training data distributions are stationary and i.i.d., and gradient descent operates under relatively stable conditions. However, in non-stationary settings, such as RL, gradient-based optimization faces severe challenges that scaling alone may exacerbate [Ceron et al., 2024a,b].

In this section, we diagnose how gradient pathologies emerge and intensify across different settings, with a focus on architectural scaling in width and depth (network scales used specified in Table 1).

## 3.1 Gradient Pathologies

We train neural networks of varying depths and widths and analyze their training dynamics.

**Supervised Learning (Stationary and Non-Stationary)** We use the CIFAR-10 image classification benchmark [Krizhevsky et al., 2009], where the input-output mapping remains fixed over time. Models consist of standard 6-layer convolutional neural networks (CNN) followed by a multi-layer perceptron (MLP). We vary the depth and width of the MLP to explore how model scale influences learning behavior. To introduce non-stationarity, we periodically shuffle the training labels during training, following the setup by Sokar et al. [2023]. This creates a loss landscape that changes over time, echoing the challenges of deep RL. Fig. 1 illustrates the contrast in training behavior and gradient flow between stationary and non-stationary supervised learning. Under non-stationarity, deep networks fail to recover accuracy, which aligns with a marked degradation in gradient magnitudes.

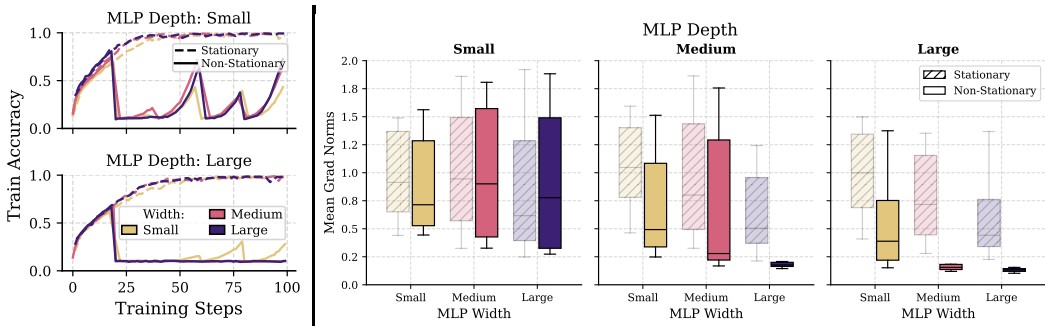

Figure 1: **Training dynamics under stationary and non-stationary supervised learning.** (Left) In the stationary setting, both shallow and deep models fit the data effectively across widths. Under non-stationarity only shallow networks partially recover during training, while deeper ones collapse. (Right) This collapse correlates with degraded gradient flow. In stationary settings, gradient norms remains stable across all network scales (shaded boxes) while in non-stationary settings (solid-colored boxes), gradient magnitudes diminish with depth and width, suggesting poor adaptability.

**Reinforcement Learning** As discussed in Sec. 2, RL introduces fundamentally different sources of non-stationarity due to the policy-dependent data distribution and moving target estimates. To study gradient dynamics, we use PQN [Gallici et al., 2025], a recent value-based algorithm that achieves strong performance without relying on a target network or replay buffer. PQN ensures stability and convergence using Layer Normalization [Ba et al., 2016] and supports GPU-based training through vectorized environments for online parallel data collection. In Sec. C.1 we extend our investigation to DQN [Mnih et al., 2015] and Rainbow [Hessel et al., 2018], demonstrating the generality of our observations. As shown in Fig. 2, deeper networks trained with PQN exhibit a collapse in both episode returns and gradient norms[2], highlighting the fragility of deep models under non-stationarity.

## 3.2 Training Degradation

In Fig. 3 we evaluate diagnostic metrics capturing expressivity and training dynamics, revealing that deeper networks exhibit pronounced training pathologies and degraded performance. We first measure the fraction of *dormant neurons*, defined as units with near-zero activations over a batch of trajectories [Sokar et al., 2023], and find that dormant neurons grow with depth, signaling underutilized capacity. Next, we assess representational expressivity using *SRank*, the effective rank of penultimate-layer activations [Kumar et al., 2020], observing that deeper networks tend to collapse state representations into lower-dimensional, and less expressive (as evidenced by declining returns) subspaces.To study loss curvature, we compute the Hessian trace of the temporal-difference loss. This metric serves as a proxy for sharpness or smoothness in optimization [Ghorbani et al., 2019], similarly to tracking the largest eigenvalue. Fig. 3 shows that only shallow networks exhibit high Hessian trace values,

---

[2]Unless otherwise specified, all ALE results are averaged over three seeds.

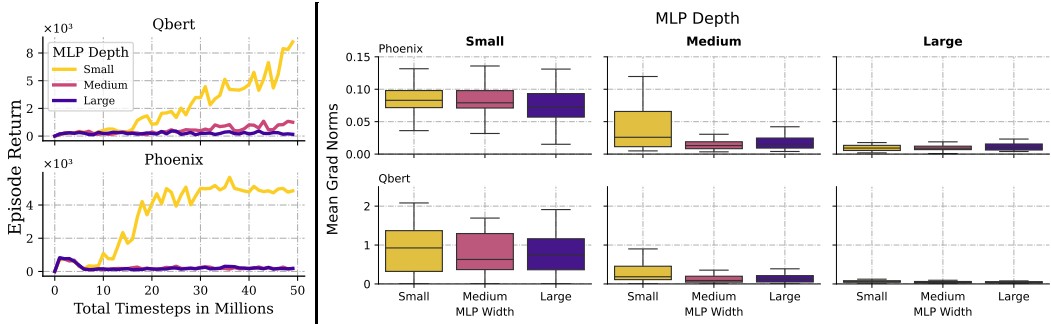

Figure 2: **Mean episode returns and gradient norms across increasing MLP depths and widths** on two ALE games using PQN. (Left) Only shallow networks achieve high episode returns; performance collapses for deeper networks. (Right) The collapse correlates with vanishing gradient norms, suggesting that deeper models fail to adapt to non-stationarity in deep RL.

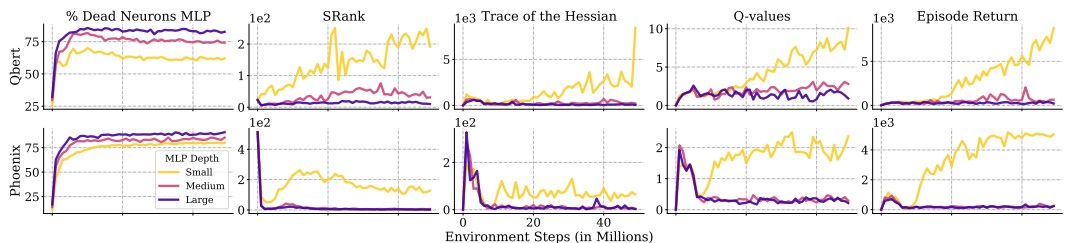

Figure 3: **Training pathologies emerge as MLP depth increases.** Deeper networks exhibit a higher fraction of inactive neurons, reduced representation rank (SRank), vanishing Hessian trace (loss curvature), and degraded learning performance (mean Q-values and episode returns). These trends indicate that scaling depth limits expressivity and plasticity, impairing policy quality.

suggesting access to sharper regions of the loss surface with pronounced directions of improvement. In contrast, deeper architectures consistently show near-zero trace, indicating poorly conditioned geometry that hinders effective gradient-based updates. These findings suggest a breakdown in representation, plasticity, and optimization as networks scale, ultimately impeding learning.

> **Key observations on gradients under non-stationarity and scale:**
> - Non-stationarity amplifies gradient degradation in deeper and wider networks.
> - In deep RL, deeper models suffer from vanishing gradients, reduced activations, and loss of representational expressivity.
> - The flat loss curvature intensifies with depth, correlating with poor learning.

## 4 Stabilizing Gradients

Having identified the pathologies that emerge in non-stationary regimes, particularly under large-scale architectures, we investigate strategies to mitigate these instabilities. We focus on two complementary interventions: skip connections [He et al., 2016] and optimizers [Martens and Grosse, 2015], as these directly improve gradient flow. We continue to use PQN as our base RL algorithm and evaluate on the Atari-10 suite [Aitchison et al., 2023]. In Sec. 5, we demonstrate that the effectiveness of our proposed gradient interventions generalize beyond this specific algorithm and environment suite.

### 4.1 Intervention 1: Multi-Skip Residuals for Gradient Stability

Gradient instability in deep networks is often aggravated by increasing depth, non-linear activations, and misaligned curvature across layers. While standard residual connections offer some relief by introducing shortcut paths for gradient flow [He et al., 2016], they typically span only one or two layers, which can be insufficient in the presence of severe gradient disruption due to non-stationarity.

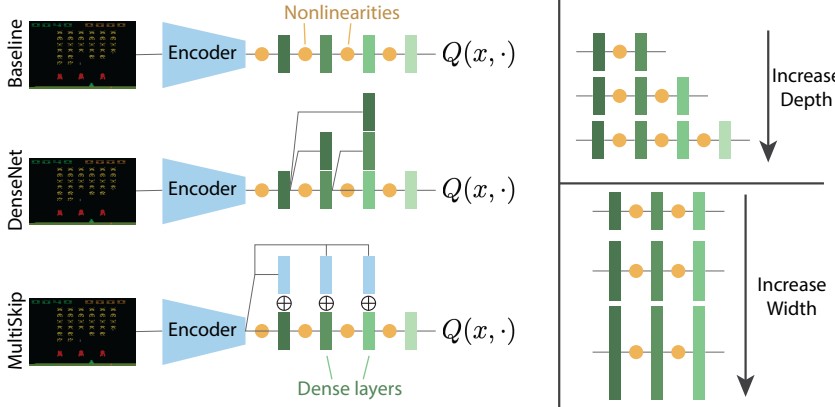

Figure 4: **(Left)** MLP architectures and **(right)** scaling strategies studied.

We introduce *multi-skip residual connections*, in which the flattened convolutional features are broadcast directly to all subsequent MLP layers. This design ensures that gradients can propagate from any depth back to the shared encoder without obstruction.

We compare our network architecture against the standard fully connected baseline across varying depths. As shown in Fig. 5 (left), performance collapses with increased depth in the baseline, while the multi-skip architecture maintains stable learning and continues to improve across widths. This improvement is accompanied by consistently higher gradient magnitudes. Complete results across all network depths and widths are presented in Sec. C.4.

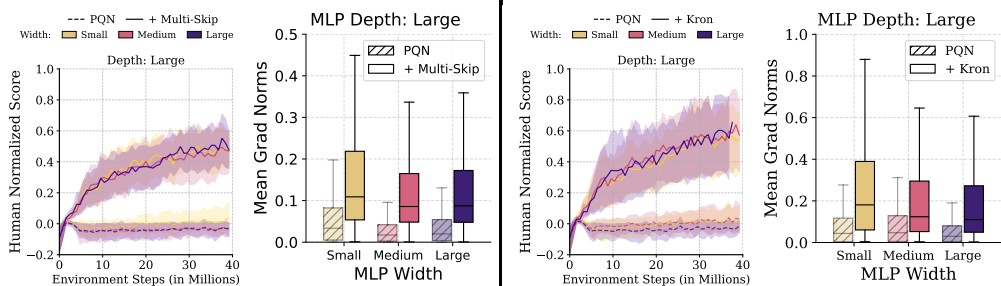

Figure 5: **Gradient-stabilizing interventions improve scalability in deep RL.** (Left) Standard fully connected networks trained with PQN collapse at greater depths due to vanishing gradients. In contrast, multi-skip architectures maintain gradient flow and scale effectively. (Right) The default RAdam optimizer leads to instability in deep networks, while switching to the Kron optimizer preserves gradient signal and enables stable learning without architectural changes.

## 4.2 Intervention 2: Second-Order Optimizers for Non-Stationarity

First-order optimizers such as SGD and Adam rely on local gradient estimates and fixed heuristics (e.g., momentum, adaptive step sizes) [Kingma and Ba, 2014], which are agnostic to curvature and often brittle under shifting data distributions. In contrast, second-order methods adjust parameter updates using curvature information, enabling more informed and stable adaptation.

Let $\mathcal{L}(\theta)$ denote the loss function, and $g = \nabla \mathcal{L}(\theta)$ its gradient. A second-order update takes the form $\theta_{t+1} = \theta_t - \eta H^{-1} g$, where $H$ is the curvature matrix, typically the Hessian or the Fisher Information Matrix (FIM) [Martens, 2020]. Directly inverting $H$ is computationally infeasible in deep neural networks so Kronecker-factored approximations, such as K-FAC [Martens and Grosse, 2015], address this challenge by approximating $H$ using low-rank Kronecker products.

Kronecker-factored optimizer (Kron for short) approximates the FIM and applies structured preconditioning that captures inter-parameter dependencies, unlike Adam's diagonal scaling. This yields directionally aware preconditioning that better aligns with the curvature of the loss surface [Martens, 2020]. In non-stationary settings, such as deep RL, where both the data distribution and curvature

evolve over time, curvature-aware updates can help preserve gradient signal by maintaining stable update magnitudes and directions. As shown in Fig. 5 (right), replacing RAdam with Kron prevents performance collapse at greater depths, even in standard MLP architectures. Complete results across all network depths and widths are presented in Sec. C.6.

## 4.3 Combining Gradient Interventions

We combine both gradient interventions to PQN and evaluate it on the full ALE suite (57 games), across 3 seeds and 200M frames. Fig. 6 shows that our augmented agent outperforms the baseline in 90% of the environments, achieving a median relative improvement of 83.27%. Notably, the baseline PQN is itself competitive with strong agents such as Rainbow [Gallici et al., 2025], highlighting the effectiveness of our interventions. Detailed per-environment learning curves can be found in Sec. E.1.

In Fig. 7 we validate the effectiveness of the combined gradient interventions in the non-stationary SL setting we used as motivation in Sec. 3. The results verify that these interventions enable high accuracy and sustained adaptability across depths and widths, even under dynamic label reshuffling.

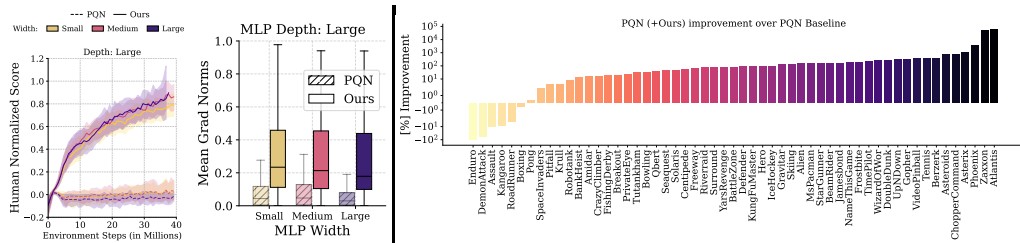

Figure 6: **Gradient-stabilized PQN achieves superior scalability.** (Left) On Atari-10, the combined interventions lead to high HNS even at greater depths, outperforming either intervention alone (see Fig. 5) and increased gradient gradient flow. (Right) On the full ALE suite, our agent outperforms the baseline in 90% of the games with a median performance improvement of 83.27%.

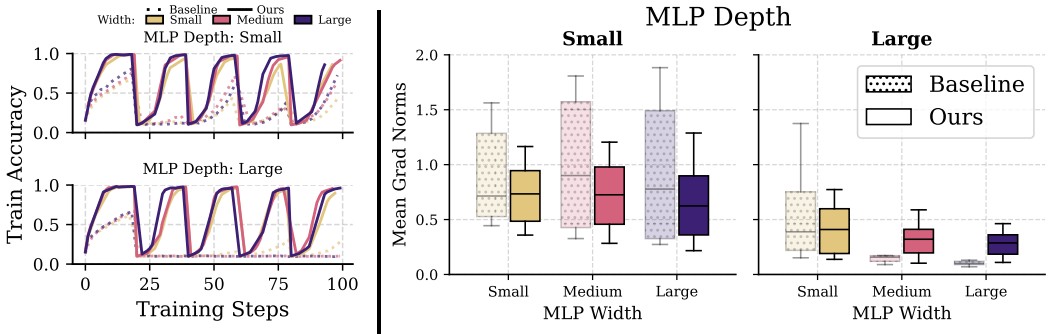

Figure 7: **Gradient interventions enable rapid recovery in non-stationary SL.** (Left) Models with combined gradient interventions rapidly recover accuracy after label reshuffling, demonstrating robust adaptation in non-stationary settings. (Right) This is supported by stable gradient flow across depth. Dashed curves and shaded boxes indicate MLP baselines.

## 4.4 Alternative Gradient-Stabilization Methods

To ensure that our findings were not specific to a narrow choice of interventions, we conducted a broader exploration of alternative strategies for improving gradient stability in deep RL. We tested a variety of approaches inspired by prior work on optimization and representation stability in both supervised and RL settings (see Sec. C.7 for more details).

As summarized in Tab. 3, none of these methods consistently improved stability or performance compared to our proposed combination of multi-skip residuals and Kronecker-factored optimization. In many cases, the alternatives yielded either negligible gains or degraded performance as network depth increased, reinforcing that architectural and curvature-aware interventions are key to preserving gradient flow at scale.

# 5 Beyond the ALE and PQN

To evaluate the generality of our findings, we extend our analyses. Specifically, we: *(i)* apply our proposed methods to PPO [Schulman et al., 2017] on the full ALE and on continuous control tasks in Isaac Gym [Makoviychuk et al., 2021]; *(ii)* assess the impact of richer convolutional encoders by replacing the standard CNN backbone used in the ALE with the Impala CNN architecture [Espeholt et al., 2018]; *(iii)* augment Simba [Lee et al., 2025] with our proposed techniques and evaluate performance on the DeepMind Control Suite (DMC) [Tassa et al., 2018]; and *(iv)* investigate whether our interventions can stabilize and scale a wide range of Q-learning algorithms in challenging offline RL benchmarks [Park et al., 2025].

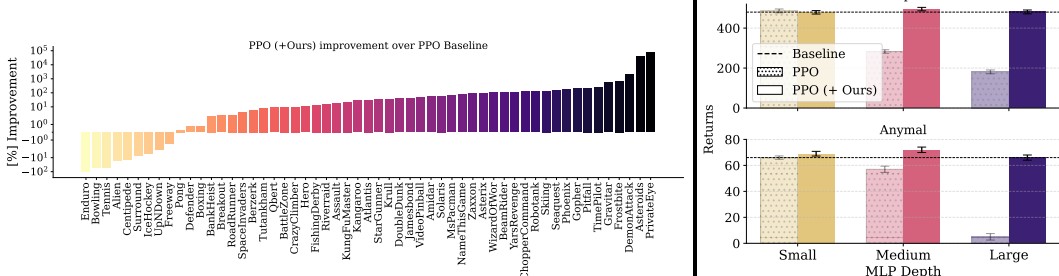

Figure 8: **PPO with gradient interventions.** Left: On the full ALE suite, applying the combined gradient interventions to PPO yields a median performance improvement of 31.40% and outperforms the baseline in 83.64% of the games. Right: In the Cartpole and Anymal tasks from IsaacGym, only the augmented PPO maintains stable performance across depths and widths.

**PPO with Gradient Interventions.** Fig. 8 (left) shows that augmenting PPO with the same strategies as in PQN (Layer Normalization by default on PQN, multi-skip residual connections, and Kronecker-factored optimization) significantly boots performance. On the ALE benchmark, the augmented PPO outperforms the baseline in 83.64% of the environments, achieving a median relative improvement of 31.40%. In Isaac Gym's continuous control tasks, including Cartpole and Anymal (Fig. 8, right), the baseline PPO collapses as model size increases, while the augmented variant remains stable and achieves superior performance at all depths and widths.

**Gradient Interventions in Scaled Encoder Variants** The Impala CNN is a scalable convolutional architecture that has demonstrated strong performance gains in agents such as Impala [Espeholt et al., 2018] and Rainbow [Hessel et al., 2018]. We investigate whether, given its capacity to extract richer representations from visual input, combining Impala CNN with our gradient flow interventions enables effective scaling of the MLP component. As shown in Fig. 9, PPO and PQN benefit significantly from replacing the standard CNN with the Impala CNN. For PQN, the Impala encoder enables successful scaling of the MLP, in contrast to the performance collapse seen without our interventions. These results suggest that the expressivity of richer visual encoders is more effectively leveraged by deeper networks when gradient flow is preserved.

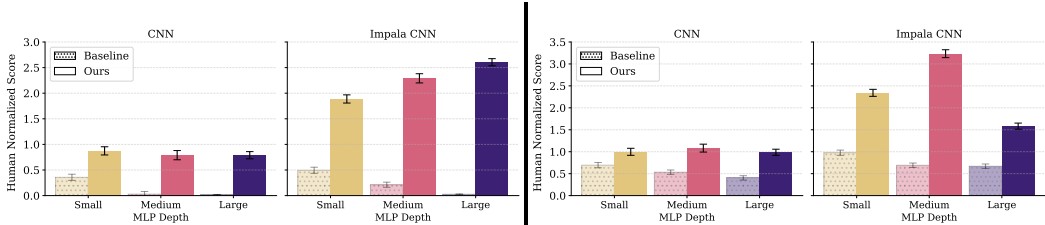

Figure 9: **Scaling performance with standard vs. Impala CNN encoders** on PQN (left) and PPO (right). Each agent is evaluated using both the Atari CNN (left sub-panels) and the Impala CNN (right sub-panels) as the encoder. Gradient interventions enable successful scaling in both cases.

**Simba with Kron Optimizer.** Simba [Lee et al., 2025] is a scalable actor-critic framework that integrates normalization, residual connections, and LayerNorm. We augment Simba by replacing its

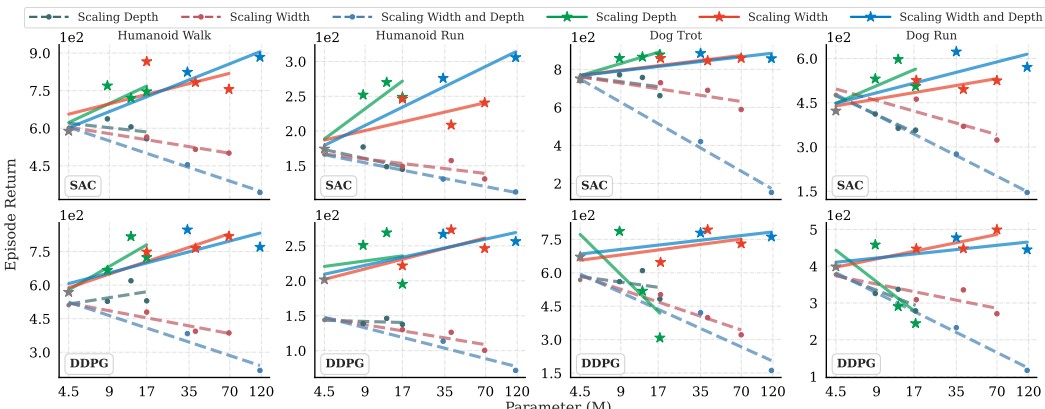

Figure 10: **Performance comparison between AdamW (dashed lines) and Kron (solid lines) optimizers using the SimBa architecture with SAC and DDPG,** averaged over 5 random seeds. As model size increases, AdamW leads to consistent performance degradation, while Kron enables stable and improved learning with larger networks.

default AdamW optimizer with Kron while keeping all other hyperparameters fixed. We evaluate SAC [Haarnoja et al., 2018] and DDPG [Lillicrap et al., 2015] on challenging DMC tasks, using the Simba architectures of varying depth and width. Despite its design for scalability, default Simba collapses across all tasks as networks grow as shown in Fig. 10 (additional results in Sec. E.2). In contrast, the Kron-augmented version successfully scales in both depth and width, achieving consistent and stable performance gains. These findings underscore the generality of our approach as effectively enabling parameter scaling in deep RL agents.

**Gradient Interventions for Scalable Offline Q-Learning.** Park et al. [2025] highlight significant challenges in scaling Q-learning algorithms for complex offline RL tasks, demonstrating that many standard offline RL baselines fail to learn effective policies, even on large, high-quality datasets. Their key finding was that performance improvements were primarily driven by techniques that shorten the effective credit assignment horizon, such as n-step returns and hierarchical methods. This led to their proposal of two new high-performing algorithms, SHARSA and DSHARSA, which are designed to operate with shorter effective horizons. This finding motivates a parallel investigation: can our proposed gradient interventions, which are designed to stabilize and accelerate deep network training, also address the scaling limitations of offline Q-learning? To test this, we augment the full suite of baselines and novel algorithms from Park et al. [2025] with our proposed gradient interventions. The results, presented in Fig. 11, show that our methods provide a complementary path to scalability. Applying our interventions generally improves the performance of the baselines across all tasks.

The performance gains are particularly pronounced in the most sparse-reward task, `humanoidmaze-giant-navigate`, where our gradient interventions enable multiple methods to achieve near-optimal performance, whereas their baseline counterparts largely fail. Furthermore, this stability extends to generalization across task difficulty. When moving from `puzzle-4x5-play` to the harder `puzzle-4x6-play` task, many baselines exhibit a sharp performance degradation. In contrast, the performance of several algorithms with our interventions remains consistent and high, demonstrating improved robustness. Finally, we note that while the primary focus of this paper is to address gradient pathologies arising from scaling and non-stationarity, these results highlight that our interventions are also highly beneficial in offline deep RL, where inputs are stationary.

## 6 Related Work

A central challenge in scaling deep RL lies in the inefficient use of model capacity. Increasing parameter counts often fails to yield proportional gains due to under-utilization. Sokar et al. [2023] and Liu et al. [2025b] show that online RL induces a growing fraction of inactive neurons, a phenomenon also observed in offline settings. Ceron et al. [2024a] report that up to 95% of parameters can be pruned post-training with negligible performance drop, underscoring substantial redundancy. These findings have motivated techniques such as weight resetting [Schwarzer et al., 2023], tokenized computation [Sokar et al., 2025], and sparse architectures [Ceron et al., 2024b, Willi et al., 2024, Liu

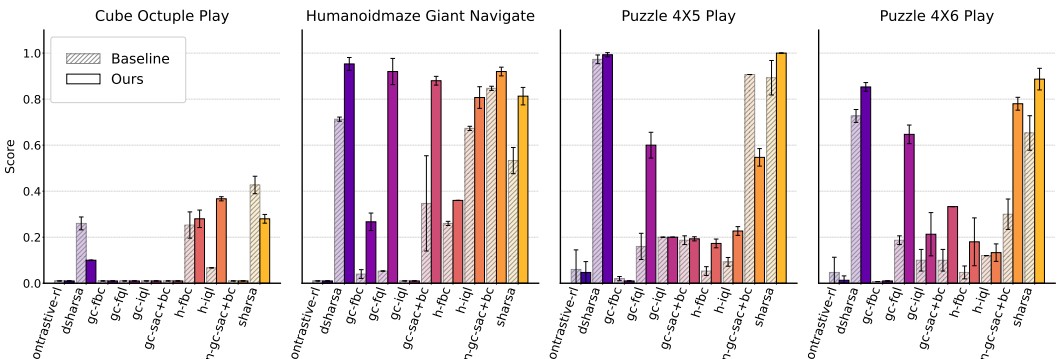

Figure 11: **Performance of offline Q-learning algorithms with and without our gradient interventions.** We compare the original algorithms from Park et al. [2025] against our augmented versions. The results, averaged over 3 seeds demonstrate a general improvement in scalability.

et al., 2025a, Ma et al., 2025], along with auxiliary objectives to promote capacity utilization [Farebrother et al., 2023]. While scaling model size offers greater expressivity, its benefits depend on appropriate training strategies [Ota et al., 2021]. Architectural interventions such as SimBa [Lee et al., 2025] improve robustness by regularizing signal propagation through components such as observation normalization, residual feedforward blocks, and layer normalization. Complementarily, BRO [Nauman et al., 2024] shows that scaling the critic network yields substantial gains in sample and compute efficiency, provided it is paired with strong regularization and optimistic exploration strategies.

Gradient flow, however, remains a central bottleneck. We complement prior efforts by explicitly targeting vanishing gradients as a mechanism for improving scalability. Our approach builds on the role of LayerNorm in stabilizing training and enhancing plasticity [Lyle et al., 2024], and leverages its theoretical effect on gradient preservation as formalized in PQN [Gallici et al., 2025]. Optimization-level interventions such as second-order methods [Martens and Grosse, 2015, Muppidi et al., 2024] and adaptive optimizers [Ellis et al., 2024, Bengio et al., 2021, Wu et al., 2017] also address instability under non-stationarity. Our approach integrates architectural and optimizer-level interventions to enable stable gradient flow and unlock parameter scaling in deep RL agents.

## 7   Discussion

Our analyses in Sec. 3 suggest that the difficulty in scaling networks in deep RL stems from the interaction between inherent non-stationarity and gradient pathologies that worsen with network size. In Sec. 4, we introduced targeted interventions to address these challenges, and in Sec. 4.3, we demonstrated their effectiveness. We validated the generality of our approach across agents and environment suites, consistently observing similar trends. These findings reaffirm the critical role of network design and optimization dynamics in training scalable RL agents. While our proposed solutions may not be optimal, they establish a strong baseline and provide a foundation for future work on gradient stabilization in deep RL. More broadly, our findings suggest that scaling limitations in deep RL are not solely attributable to algorithmic instability or insufficient exploration, but also stem from gradient pathologies amplified by architectural and optimization choices. Addressing these issues directly, without altering the learning algorithm, yields substantial gains in scalability and performance. This suggests that ensuring stable gradient flow is a necessary precondition for effective parameter scaling in deep RL.

**Limitations.** Our study is constrained by computational resources, which limited our ability to explore architectures beyond a certain size. While our interventions show consistent improvements across agents and environments, further scaling remains an open question. While using second order optimizers introduced additional computational overhead (see Tab. 12), this cost is mitigated by leveraging vectorized environments and efficient deep RL algorithms, narrowing the gap relative to standard methods. These limitations highlight promising directions for future work, including the development of more computationally efficient gradient stabilization strategies and scalable optimization techniques.

**Acknowledgment**    The authors would like to thank João Guilherme Madeira Araújo, Evan Walters, Olya Mastikhina, Dhruv Sreenivas, Ali Saheb Pasand, Ayoub Echchahed and Gandharv Patil for valuable discussions during the preparation of this work. João Araújo deserves a special mention for providing us valuable feed-back on an early draft of the paper. We want to acknowledge funding support from Natural Sciences and Engineering Research Council (NSERC) of Canada, Google Research, Fonds de recherche du Québec (FRQNT) and The Canadian Institute for Advanced Research (CIFAR) and compute support from Digital Research Alliance of Canada, Mila IDT, and NVidia. We would also like to thank the Python community Van Rossum and Drake Jr [1995], Oliphant [2007] for developing tools that enabled this work, including NumPy Harris et al. [2020], Matplotlib Hunter [2007], Jupyter Kluyver et al. [2016], and Pandas McKinney [2013].

**Broader Impact**    This paper presents work whose goal is to advance the field of Machine Learning. There are many potential societal consequences of our work, none which we feel must be specifically highlighted here.

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

## A  Environment Details

Throughout the paper, we evaluate the deep reinforcement learning agents' performance on the Atari-10 suite [Aitchison et al., 2023], a curated subset of games from the Arcade Learning Environment (ALE) [Bellemare et al., 2013]. Atari-10 consists of 10 games selected to capture the maximum variance in algorithm performance, achieving over 90% correlation with results on the full ALE benchmark. This makes it a computationally efficient yet representative testbed for deep reinforcement learning. We follow the experimental protocol of Obando Ceron et al. [2023], Ceron et al. [2024b], Agarwal et al. [2021], running each experiment with three random seeds and reporting the aggregate human-normalized score across games.

The games in Atari-10 are:

- Amidar, Battle Zone, Bowling, Double Dunk, Frostbite, Kung Fu Master, Name This Game, Phoenix, Q*Bert and River Raid.

Additionally, to further support the generality of our findings, we evaluate the proposed combined gradient interventions on the full ALE benchmark. We also assess their effectiveness on continuous control tasks from the IsaacGym simulator [Makoviychuk et al., 2021] and the DeepMind Control Suite (DMC) [Tassa et al., 2018], extending our analysis to robotics-based environments. We conduct experiments on the 4 challenging tasks of DMC:

- Humanoid Walk, Humanoid Run, Dog Trot and Dog Run.

## B  Network Sizes

Throughout the paper, we experiment with models of varying depths and widths. Unless stated otherwise (e.g. in Sec. 5, where we evaluate the Impala CNN), the convolutional feature extractors are kept fixed. Consequently, our experiments focus primarily on scaling strategies and architectural variations in the MLP components of the networks.

To enable meaningful comparisons across different learning regimes, the MLP architectures are kept consistent across supervised learning (SL), non-stationary SL, and reinforcement learning (RL) experiments. This consistency ensures that observed differences in gradient behavior arise from the learning setting itself, rather than confounding factors due to domain-specific architectures.

Table 1 provides detailed information on the number of parameters for each depth–width configuration, categorized as small, medium, or large, as used throughout the paper.

Table 1: Number of parameters (in millions) for different MLP architectures.

| Depth / Width | Small | Medium | Large |
|---|---|---|---|
| Small | 2.39 | 11.90 | 27.70 |
| Medium | 3.45 | 21.35 | 53.93 |
| Large | 4.50 | 30.79 | 80.15 |

## C  Additional Experiments

### C.1  Scaling with DQN and Rainbow

To further support our hypothesis on the emergence of gradient pathologies in deep reinforcement learning, we investigate whether similar issues arise in algorithms beyond PQN and PPO, as discussed in the main paper. Specifically, we study the effects of architectural scaling on two widely used value-based algorithms: DQN [Mnih et al., 2015] and Rainbow [Hessel et al., 2018].

DQN is a foundational deep RL algorithm that learns action-value functions using temporal difference updates and experience replay, serving as a standard baseline for value-based methods. Rainbow extends DQN by integrating several enhancements, such as double Q-learning, prioritized experience

replay, dueling networks, multi-step learning, distributional value functions, and noisy exploration, to achieve improved sample efficiency and stability.

In Fig. 12, we report the performance of DQN and Rainbow as we scale the depth and width of their networks. As with PQN and PPO, we observe consistent degradation in performance at larger scales. In Fig. 13, we present the corresponding gradient behavior, which reveals the same vanishing and destabilization phenomena discussed in this work. These findings reinforce the generality of the identified gradient pathologies across both policy-based and value-based deep RL algorithms.

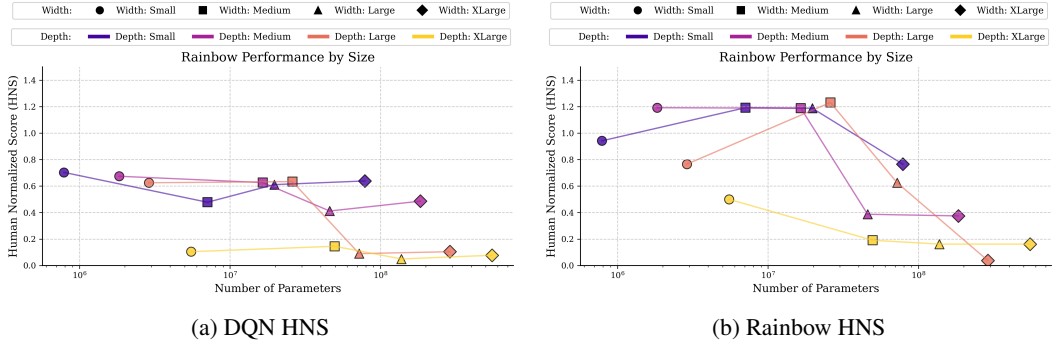

(a) DQN HNS

(b) Rainbow HNS

Figure 12: **Median human normalized scores for DQN (left) and Rainbow (right) as a function of total network parameters.** Lines of different colors denote varying network depths, while marker shapes indicate different widths. For both agents, performance consistently declines as network size increases, highlighting the adverse effects of scaling.

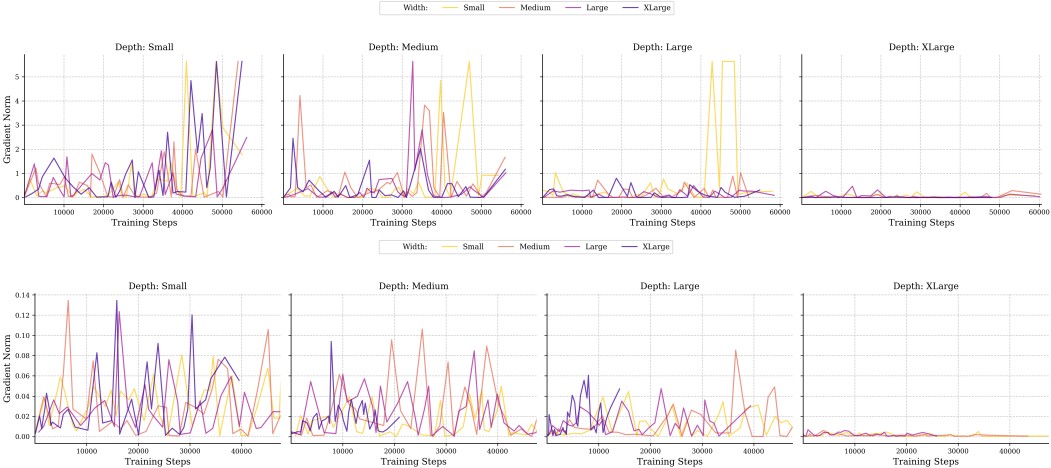

Figure 13: **Gradient magnitudes during training for DQN (top) and Rainbow (bottom).** As network depth increases, gradient flow systematically diminishes, ultimately collapsing to near-zero values. This consistent decay mirrors the performance degradation observed at larger scales.

## C.2 Combining Gradient Interventions in Non-stationary Supervised Learning

Building on our findings in Sec. 4.3, we extend our analysis by applying the proposed combined gradient interventions to the same image classification models used in Sec. 3. Specifically, we train the models in the non-stationary supervised learning setup, where the CIFAR-10 labels are iteratively shuffled, following the experimental design from Sokar et al. [2023]. As demonstrated in Sec. 3, while models in standard supervised learning settings are able to scale effectively and maintain high performance, introducing non-stationarity leads to failure in adaptation for baselines that use fully connected layers and the Adam optimizer. This issue is exacerbated as model scale increases.

Our results, presented in Fig. 7, show that combining the multi-skip architecture for the MLP component with the Kronecker-factored optimizer and Layer Normalization enables near-perfect

continuous adaptation. The models quickly adapt to the changing optimization problem following label reshuffling, with gradient magnitudes remaining stable throughout the process.

## C.3 Architecture and Optimizer Ablations

In this work, we introduce the multi-skip architecture, an extension of the standard residual MLP design, and propose the use of the Kronecker-factored optimizer for online deep RL. While these techniques form the basis of our primary interventions, our broader goal is not to prescribe a fixed set of methods, but rather to motivate a general class of architectural and optimization interventions that promote healthy gradient flow in deep networks. To this end, we expand the scope of our evaluation by incorporating a wider range of baselines. Specifically, we compare various optimizer choices, including Adam and AdaBelief [Zhuang et al., 2020], alongside MLP architectures such as the standard residuapl MLP [He et al., 2016] and DenseNet [Huang et al., 2017]. These architectures have been previously explored in the context of scaling networks in online deep RL [Lee et al., 2025, Ota et al., 2024], providing a relevant basis for comparison.

We also evaluated state-of-the-art optimizers that have demonstrated success in training large-scale models such as transformers in supervised learning. Specifically, we tested Shampoo [Gupta et al., 2018], a second-order optimizer that maintains and preconditions gradients using full-matrix statistics per layer, and Apollo [Ma, 2020], an adaptive optimizer that leverages curvature information without explicitly computing or storing second-order matrices.

Despite extensive hyperparameter tuning for both methods, we were unable to achieve strong performance in the online deep RL setting. This suggests that further investigation is needed to understand the key properties required for these optimizers to be effective in this regime. Asadi et al. [2023], Ceron and Castro [2021] demonstrate that optimizer behavior plays a critical role in the training dynamics of online deep RL methods, with Asadi et al. [2023] showing that stale optimizer states can hinder learning, and Ceron and Castro [2021] revealing that optimizer sensitivity interacts with the choice of loss function, particularly when comparing Huber and MSE losses.

We present the results for PPO and PQN across all tested optimizers in Fig. 14.

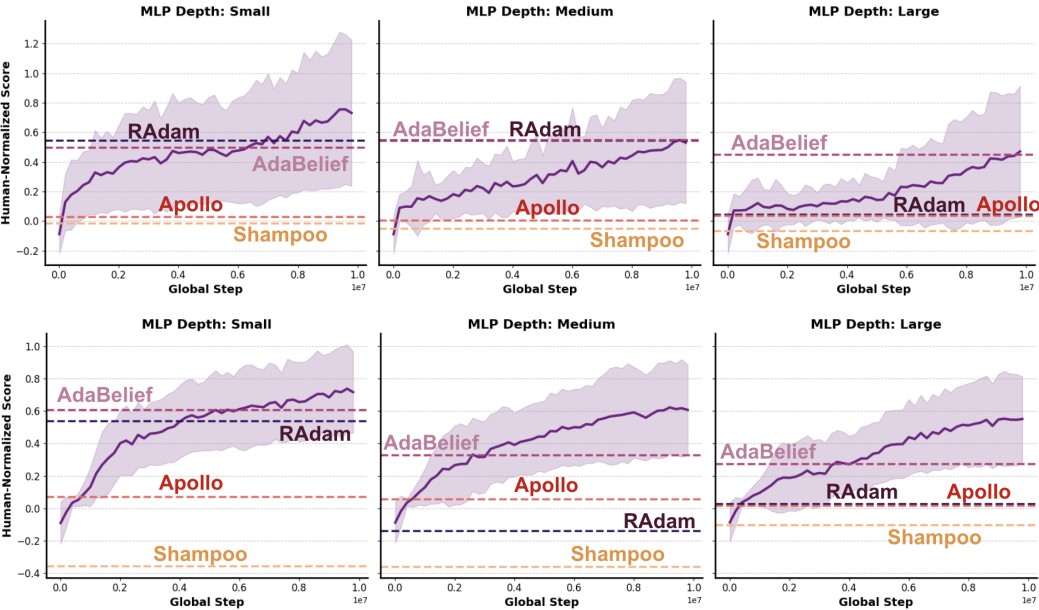

Figure 14: **Median human normalized scores on Atari-10 for PPO (top row) and PQN (bottom row), comparing a range of optimizers including RAdam, AdaBelief, Shampoo, Apollo, and Kron (shown in the main curves).** While adaptive optimizers like AdaBelief show some robustness, only Kron consistently enables stable and performant training as models scale. Each curve represents the mean performance across three random seeds per algorithm, with shaded areas indicating 95% bootstrap confidence intervals.

## C.4 Results with the Multi-Skip Architecture.

We present the full learning curves comparing the proposed multi-skip architecture to the baseline fully connected architecture across all depths and widths studied in the paper. We follow the experimental protocol of Obando Ceron et al. [2023], Ceron et al. [2024b], Agarwal et al. [2021], running each experiment with three random seeds.

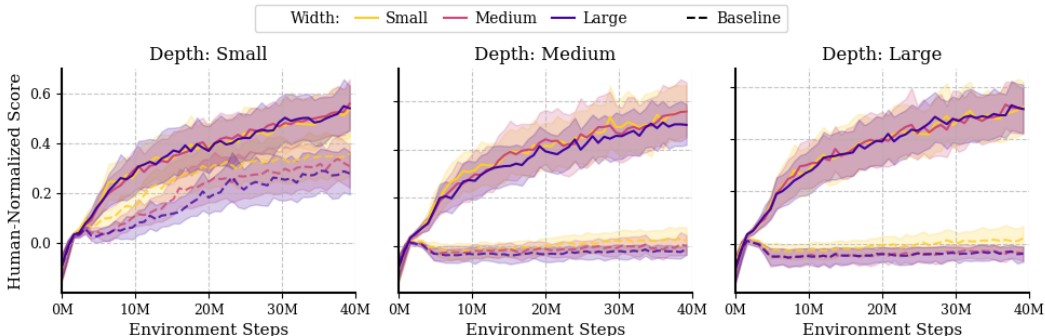

Figure 15: **Median human-normalized scores with PQN on the Atari-10 benchmark, comparing the baseline agent and the proposed multi-skip architecture across varying depths and widths.** The multi-skip architecture not only improves performance at shallow depths, but also enables PQN to remain trainable across all scales considered, whereas the baseline MLP rapidly collapses as depth and width increase. Each curve represents the mean performance across three random seeds per algorithm, with shaded areas indicating 95% bootstrap confidence intervals.

## C.5 Ablation on the number of skip connections.

To isolate the effect of skip length on performance, we fix the main network of our proposed MultiSkip architecture (in *large* size, which includes 5 residual blocks) and vary how many of these blocks receive skip connections from the encoder. When Skip = $k$, we apply the encoder features as skip connections to the first $k$ residual blocks immediately following the encoder, while the remaining $(5 - k)$ blocks operate without direct encoder input. The table below reports human-normalized scores on the Atari-10 benchmark Agarwal et al. [2021].

Table 2: Human-normalized scores on the Atari-10 benchmark, varying the number of residual blocks ($k$) that receive skip connections from the encoder. Performance generally improves as more connections are added.

| Environment | Skip=1 | Skip=2 | Skip=3 | Skip=4 | Skip=5 |
|---|---|---|---|---|---|
| Amidar-v5 | 0.20 | 0.17 | 0.19 | 0.20 | **0.36** |
| BattleZone-v5 | 0.01 | 0.67 | 0.62 | 0.60 | **0.69** |
| Bowling-v5 | 0.07 | 0.04 | 0.04 | 0.08 | **0.23** |
| DoubleDunk-v5 | -2.09 | -2.00 | -1.55 | -1.36 | **-1.32** |
| Frostbite-v5 | 0.67 | 0.70 | 0.79 | **0.92** | 0.88 |
| KungFuMaster-v5 | 0.93 | 0.95 | 0.93 | **1.12** | 0.98 |
| NameThisGame-v5 | 0.79 | 0.65 | 0.72 | 0.85 | **1.24** |
| Phoenix-v5 | **0.69** | 0.68 | 0.68 | 0.66 | 0.66 |
| Qbert-v5 | 0.84 | 1.01 | **1.06** | 0.91 | 1.09 |
| Riverraid-v5 | 0.42 | 0.44 | 0.65 | 0.69 | **0.99** |
| **Aggregate (mean)** | 0.25 | 0.33 | 0.41 | 0.47 | **0.58** |

Performance steadily improves as more skip connections are added, peaking when all 5 blocks are connected. This supports our original design decision to broadcast features to all MultiSkip blocks.

## C.6 Results with the Kron Optimizer.

We present the full learning curves comparing the Kron optimizer to the baseline RAdam optimizer originally used in PQN [Gallici et al., 2025], across all depths and widths studied in the paper. We follow the experimental protocol of Obando Ceron et al. [2023], Ceron et al. [2024b], Agarwal et al. [2021], running each experiment with three random seeds.

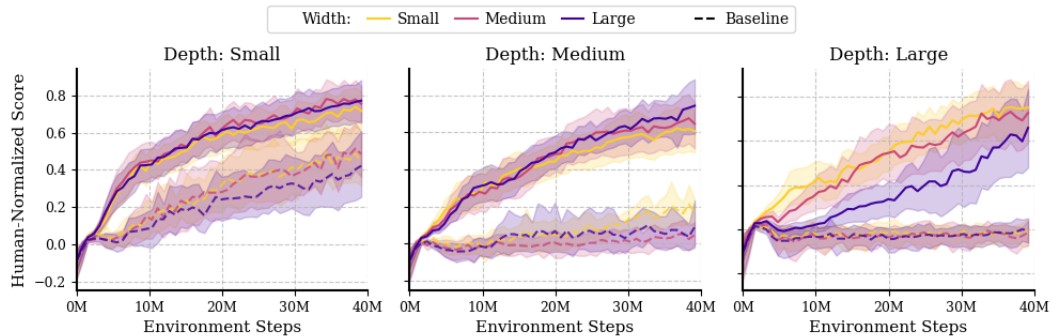

Figure 16: **Median human-normalized scores with PQN on the Atari-10 benchmark, comparing the Kron optimizer to the baseline RAdam optimizer across varying depths and widths.** Similar to the multi-skip architecture, Kron not only improves performance at shallow depths, but also enables PQN to remain trainable across all scales considered. In contrast, performance with RAdam rapidly collapses as depth and width increase. Each curve represents the mean performance across thee random seeds per algorithm, with shaded areas indicating 95% bootstrap confidence intervals.

## C.7 Justification for Selected Interventions.

The choice of Kronecker-factored optimization and the MultiSkip architecture was the result of a systematic exploration of candidate interventions aimed at improving gradient flow, plasticity, and stability in deep RL. We evaluated a wide range of alternative methods from prior work known to mitigate optimization pathologies [Moalla et al., 2024, Juliani and Ash, 2024].

Our evaluated methods, summarized in Tab. 3, included:

- Second-order and adaptive optimizers: Apollo [Ma, 2020], Shampoo [Gupta et al., 2018], AdaBelief [Zhuang et al., 2020].
- Regularization: L2 norm penalties [Kumar et al., 2023], weight clipping, and weight decay [Elsayed et al., 2024].
- Activation functions: GELU [Hendrycks, 2016] and CReLU [Abbas et al., 2023].
- Learning rate schedules: Cosine annealing and cyclic schedulers.
- Learning rate scaling: Multiplying and dividing the default learning rate (2.5e-4) by 10 to compensate for increased network scale.

As shown in the table, none of these interventions consistently improved performance compared to our proposed combination. This motivated our decision to focus on the combination of Kronecker-factored optimization and the multi-skip architecture.

Table 3: **Comparison of mean human-normalized scores on Atari-10 for alternative interventions.** Our proposed method (Ours) is compared against ablations using only a single alternative intervention. The results (3 seeds per experiment) show that no single alternative consistently matches the performance of our combined approach.

| Environment | Ours | Cosine LR | Cyclic LR | GELU | CReLU | L2 Norm | Weight Clip | Weight Decay | LR=2.08e-5 | LR=3.00e-3 |
|---|---|---|---|---|---|---|---|---|---|---|
| Amidar-v5 | **0.355897** | -0.001663 | 0.002276 | 0.015143 | 0.016514 | 0.029293 | 0.012283 | 0.013013 | 0.018848 | 0.078660 |
| BattleZone-v5 | **0.694566** | 0.031297 | -0.000287 | 0.015505 | -0.004594 | -0.004594 | 0.035604 | 0.061446 | 0.019812 | 0.575407 |
| Bowling-v5 | **0.231831** | 0.050145 | -0.008721 | 0.028706 | -0.009811 | -0.000727 | 0.030523 | 0.051235 | -0.086483 | 0.067587 |
| DoubleDunk-v5 | **-1.318182** | -2.318182 | -2.318182 | -2.363636 | -2.318182 | -2.363636 | -2.454545 | -2.454545 | -2.454545 | -2.090909 |
| Frostbite-v5 | **0.881087** | -0.001921 | 0.027708 | 0.008890 | -0.010938 | 0.004755 | 0.075372 | 0.021736 | 0.008034 | 0.110036 |
| KungFuMaster-v5 | 0.975251 | -0.011055 | 0.003181 | -0.000823 | -0.011500 | -0.011278 | -0.011500 | -0.011055 | -0.011055 | **1.362077** |
| NameThisGame-v5 | **1.242066** | -0.030191 | -0.132767 | -0.261400 | -0.198430 | -0.068147 | -0.197995 | -0.148227 | -0.134417 | 0.747251 |
| Phoenix-v5 | **0.655141** | 0.072687 | -0.087237 | -0.090014 | -0.090168 | -0.068722 | -0.090940 | -0.093949 | -0.102358 | 0.411968 |
| Qbert-v5 | **1.094894** | 0.013814 | 0.010052 | -0.002833 | 0.008359 | 0.006102 | 0.016259 | 0.000177 | 0.001305 | 0.112282 |
| Riverraid-v5 | **0.993568** | -0.000127 | 0.345100 | 0.304636 | -0.056846 | -0.058494 | 0.293514 | -0.044456 | 0.057765 | 0.237365 |

# D  Hyper-parameters

Below, we provide details of the hyperparameters used throughout the paper for each algorithm. In general, they match those proposed in the corresponding original papers.

Table 4: PQN Hyperparameters

| Hyperparameter | Value / Description |
|---|---|
| Learning rate | 2.5e-4 |
| Anneal lr | False (no learning rate annealing) |
| Num envs | 128 (parallel environments) |
| Num steps | 32 (steps per rollout per environment) |
| Gamma | 0.99 (discount factor) |
| Num minibatches | 32 |
| Update epochs | 2 (policy update epochs) |
| Max grad norm | 10.0 (gradient clipping) |
| Start e | 1.0 (initial exploration rate) |
| End e | 0.005 (final exploration rate) |
| Exploration fraction | 0.10 (exploration annealing fraction) |
| Q lambda | 0.65 ($Q(\lambda)$ parameter) |
| Use ln | True (use layer normalization) |
| Activation fn | relu (activation function) |

Table 5: PPO Hyperparameters

| Hyperparameter | Value / Description |
| --- | --- |
| Learning rate | 2.5e-4 |
| Num envs | 8 |
| Num steps | 128 (steps per rollout per environment) |
| Anneal lr | True (learning rate annealing enabled) |
| Gamma | 0.99 (discount factor) |
| Gae lambda | 0.95 (GAE parameter) |
| Num minibatches | 4 |
| Update epochs | 4 |
| Norm adv | True (normalize advantages) |
| Clip coef | 0.1 (PPO clipping coefficient) |
| Clip vloss | True (clip value loss) |
| Ent coef | 0.01 (entropy regularization coefficient) |
| Vf coef | 0.5 (value function loss coefficient) |
| Max grad norm | 0.5 (gradient clipping threshold) |
| Use ln | False (no layer normalization) |
| Activation fn | relu (activation function) |
| Shared cnn | True (shared CNN between policy and value networks) |

Table 6: PPO Hyperparameters for IsaacGym

| Hyperparameter | Value / Description |
| --- | --- |
| Total timesteps | 30,000,000 |
| Learning rate | 0.0026 |
| Num envs | 4096 (parallel environments) |
| Num steps | 16 (steps per rollout) |
| Anneal lr | False (disable learning rate annealing) |
| Gamma | 0.99 (discount factor) |
| Gae lambda | 0.95 (GAE lambda) |
| Num minibatches | 2 |
| Update epochs | 4 (update epochs per PPO iteration) |
| Norm adv | True (normalize advantages) |
| Clip coef | 0.2 (policy clipping coefficient) |
| Clip vloss | False (disable value function clipping) |
| Ent coef | 0.0 (entropy coefficient) |
| Vf coef | 2.0 (value function loss coefficient) |
| Max grad norm | 1.0 (max gradient norm) |
| Use ln | False (no layer normalization) |
| Activation fn | relu (activation function) |

Table 7: DQN Hyperparameters

| Hyperparameter | Value / Description |
| --- | --- |
| Learning rate | 1e-4 |
| Num envs | 1 |
| Buffer size | 1,000,000 (replay memory size) |
| Gamma | 0.99 (discount factor) |
| Tau | 1.0 (target network update rate) |
| Target network frequency | 1000 (timesteps per target update) |
| Batch size | 32 |
| Start e | 1.0 (initial exploration epsilon) |
| End e | 0.01 (final exploration epsilon) |
| Exploration fraction | 0.10 (fraction of total timesteps for decay) |
| Learning starts | 80,000 (timesteps before training starts) |
| Train frequency | 4 (training frequency) |
| Use ln | False (no layer normalization) |
| Activation fn | relu (activation function) |

Table 8: Rainbow Hyperparameters

| Hyperparameter | Value / Description |
| --- | --- |
| Learning rate | 6.25e-5 |
| Num envs | 1 |
| Buffer size | 1,000,000 (replay memory size) |
| Gamma | 0.99 (discount factor) |
| Tau | 1.0 (target network update rate) |
| Target network frequency | 8000 (timesteps per target update) |
| Batch size | 32 |
| Start e | 1.0 (initial exploration epsilon) |
| End e | 0.01 (final exploration epsilon) |
| Exploration fraction | 0.10 (fraction of total timesteps for decay) |
| Learning starts | 80,000 (timesteps before training starts) |
| Train frequency | 4 (training frequency) |
| N step | 3 (n-step Q-learning horizon) |
| Prioritized replay alpha | 0.5 |
| Prioritized replay beta | 0.4 |
| Prioritized replay eps | 1e-6 |
| N atoms | 51 (number of atoms in distributional RL) |
| V min | -10 (value distribution lower bound) |
| V max | 10 (value distribution upper bound) |
| Use ln | False (no layer normalization) |
| Activation fn | relu (activation function) |

Table 9: Image Classification Hyperparameters (CIFAR-10)

| Hyperparameter | Value |
| --- | --- |
| Batch size | 256 |
| Epochs | 100 |
| Learning rate | 0.00025 |

Table 10: SAC Hyperparameters

| Hyperparameter | Value / Description |
| --- | --- |
| Critic block type | SimBa |
| Critic num blocks | {2, 4, 6, 8} |
| Critic hidden dim | {512, 1024, 1536, 2048} |
| Target critic momentum ($\tau$) | 5e-3 |
| Actor block type | SimBa |
| Actor num blocks | {1, 2, 3, 4} |
| Actor hidden dim | {128, 256, 384, 512} |
| Initial temperature ($\alpha_0$) | 1e-2 |
| Temperature learning rate | 1e-4 |
| Target entropy ($\mathcal{H}^*$) | $|\mathcal{A}|/2$ |
| Batch size | 256 |
| Optimizer | {AdamW, Kron} |
| AdamW's learning rate | 1e-4 |
| Kron's learning rate | 5e-5 |
| Optimizer momentum ($\beta_1$, $\beta_2$) | (0.9, 0.999) |
| Weight decay ($\lambda$) | 1e-2 |
| Discount ($\gamma$) | Heuristic |
| Replay ratio | 2 |
| Clipped Double Q | False |

Table 11: DDPG Hyperparameters

| Hyperparameter | Value / Description |
| --- | --- |
| Critic block type | SimBa |
| Critic num blocks | {2, 4, 6, 8} |
| Critic hidden dim | {512, 1024, 1536, 2048} |
| Critic learning rate | 1e-4 |
| Target critic momentum ($\tau$) | 5e-3 |
| Actor block type | SimBa |
| Actor num blocks | {1, 2, 3, 4} |
| Actor hidden dim | {128, 256, 384, 512} |
| Actor learning rate | 1e-4 |
| Exploration noise | $\mathcal{N}(0, 0.1^2)$ |
| Batch size | 256 |
| Optimizer | {AdamW, Kron} |
| AdamW's learning rate | 1e-4 |
| Kron's learning rate | 5e-5 |
| Optimizer momentum ($\beta_1$, $\beta_2$) | (0.9, 0.999) |
| Weight decay ($\lambda$) | 1e-2 |
| Discount ($\gamma$) | Heuristic |
| Replay ratio | 2 |
| Clipped Double Q | False |

# E   Compute Details

All experiments were conducted on a single-GPU setup using an NVIDIA RTX 8000, 12 CPU workers, and 50GB of RAM.

Table 12: **Training times across model scales for two optimizers** K-FAC shows increased cost as depth and width grow.

| Depth | Width | Optimizer | Time |
|-------|-------|-----------|------|
| *RAdam* | | | |
| Small | Small | Adam | 51m |
| Small | Medium | Adam | 53m |
| Small | Large | Adam | 57m |
| Medium | Small | Adam | 1h 4m |
| Medium | Medium | Adam | 1h 10m |
| Medium | Large | Adam | 1h 11m |
| Large | Small | Adam | 1h 18m |
| Large | Medium | Adam | 1h 18m |
| Large | Large | Adam | 1h 27m |
| *Kron* | | | |
| Small | Small | Kron | 1h 59m |
| Small | Medium | Kron | 2h 27m |
| Small | Large | Kron | 3h 38m |
| Medium | Small | Kron | 2h 44m |
| Medium | Medium | Kron | 3h 32m |
| Medium | Large | Kron | 5h 59m |
| Large | Small | Kron | 3h 27m |
| Large | Medium | Kron | 4h 36m |
| Large | Large | Kron | 7h 42m |

## E.1   Results on the Full ALE

In this section, we provide the full training curves corresponding to the aggregated results shown in Sec. 4.3, where we evaluate the performance of the PQN and PPO agents on the full set of environments from the ALE after applying our two proposed gradient interventions. The per-environment learning curves are presented in Fig. 17 for PQN and Fig. 18 for PPO. We follow the experimental protocol of Obando Ceron et al. [2023], Ceron et al. [2024b], Agarwal et al. [2021], running each experiment with three random seeds.

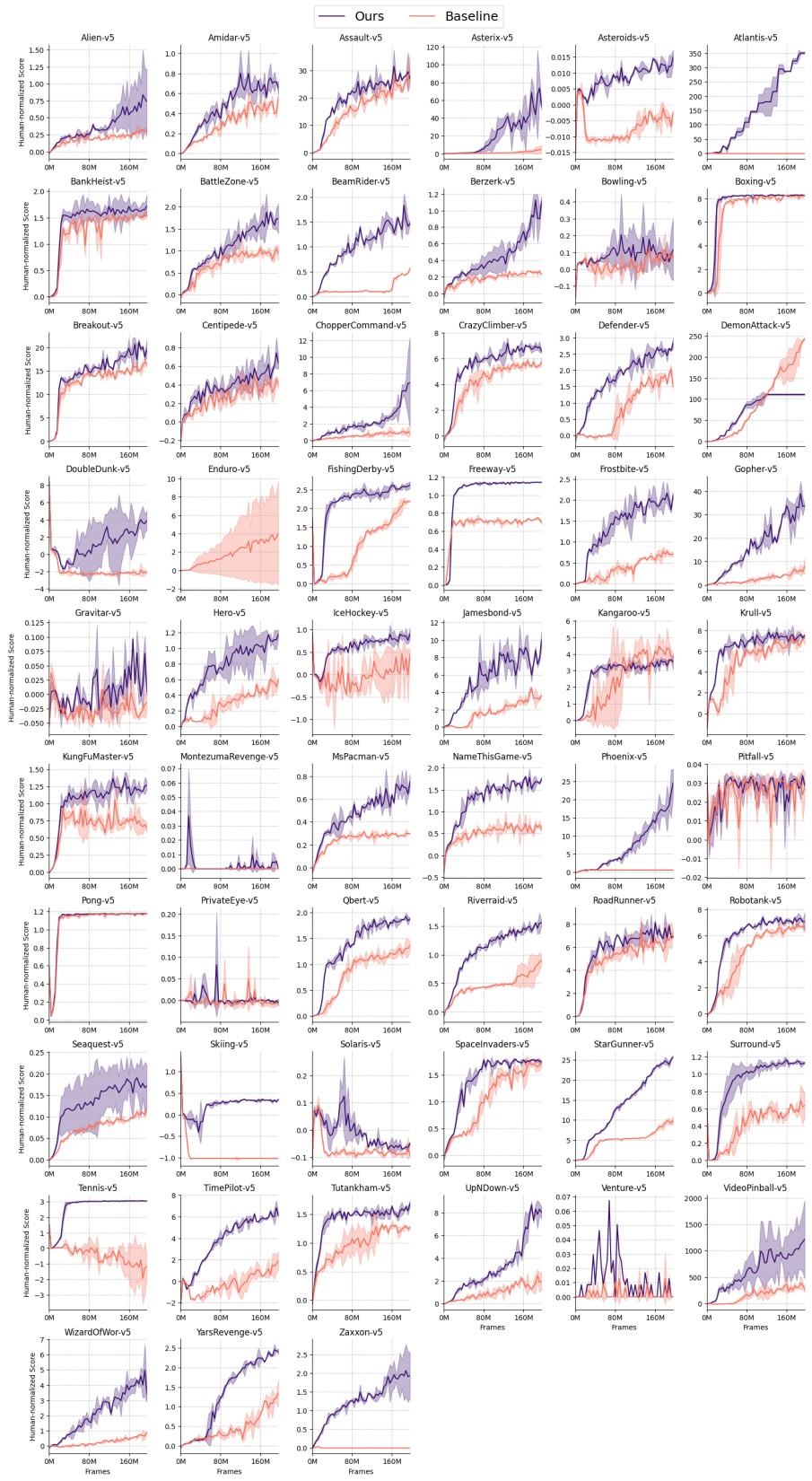

Figure 17: Mean human-normalized score on the full ALE suite, comparing the baseline PQN agent (light curves) with the augmented agent using our combined gradient interventions (dark curves).

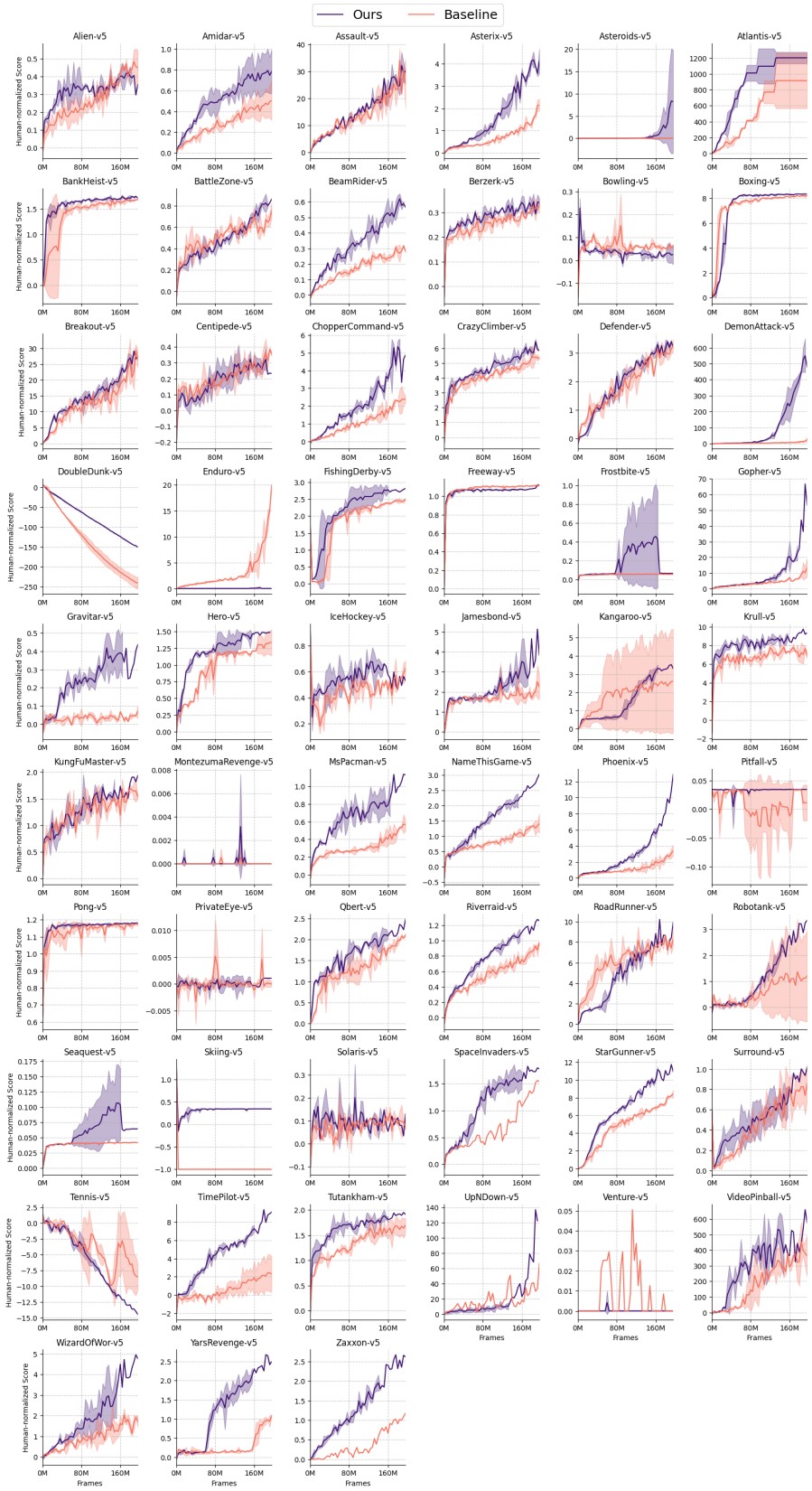

Figure 18: Mean human-normalized score on the full ALE suite, comparing the baseline PPO agent (light curves) with the augmented agent using our combined gradient interventions (dark curves).

## E.2 Simba on DMC

In this section, we present the full results accompanying the experiments combining Simba [Lee et al., 2025] with our proposed gradient interventions, as introduced in Sec. 4.3. For these experiments, we retain Simba's original architectural choices but replace the AdamW optimizer with Kron.

We compare Simba using both SAC and DDPG as the underlying RL algorithms. While SAC generally outperforms DDPG, we consistently observe that scaling depth and width, either independently or jointly, leads to a degradation in performance with Simba. However, this degradation is mitigated, and in many cases reversed, when using the Kron optimizer, resulting in improved performance as model capacity increases.

The following figures illustrate these findings:

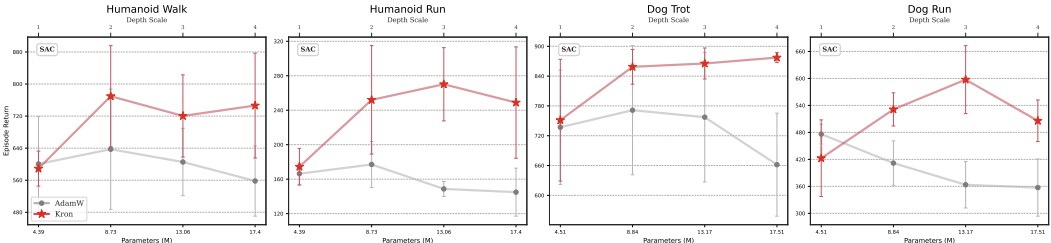

Figure 19: SAC scaling depth

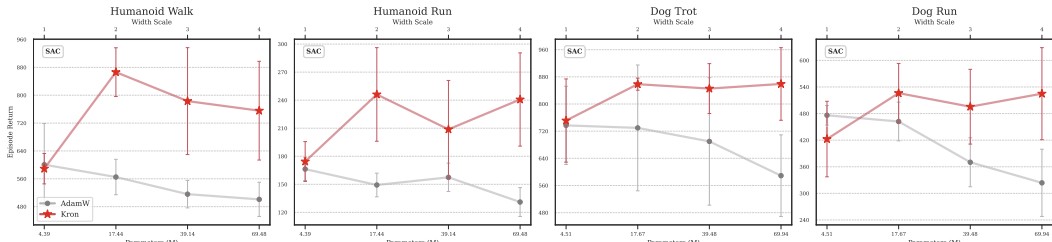

Figure 20: SAC scaling width

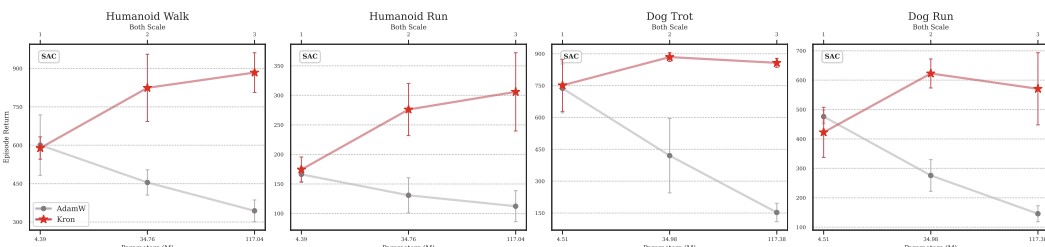

Figure 21: SAC scaling both depth and width

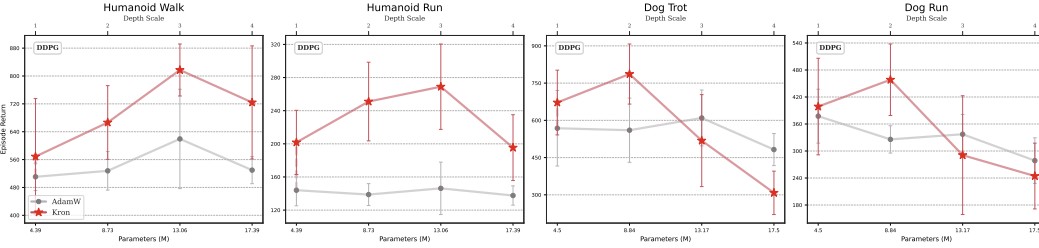

Figure 22: DDPG scaling depth

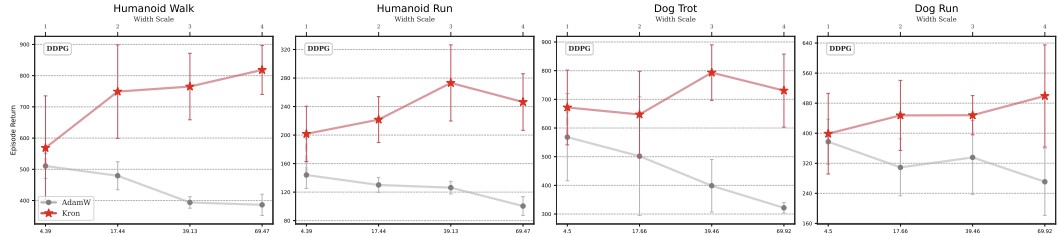

Figure 23: DDPG scaling width

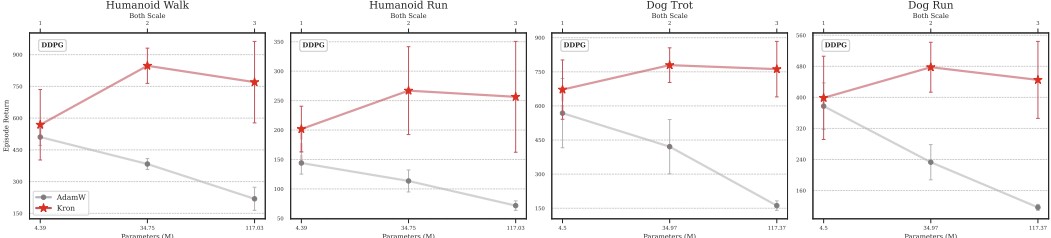

Figure 24: DDPG scaling both depth and width

