# OpenReview forum: "Stable Gradients for Stable Learning at Scale in Deep Reinforcement Learning"
_NeurIPS.cc/2025/Conference — NeurIPS 2025 spotlight_

### Official Review · Reviewer_Uxog · 2025-06-24

**Clarity:** 4
**Significance:** 3
**Originality:** 3
**Rating:** 5
**Confidence:** 4

**Summary:**

This paper investigates the scaling of network architectures in deep reinforcement learning, which often leads to degrading performance, contrary to trends observed in supervised learning. The authors provide empirical evidence on a continual learning toy task based on CIFAR10, as well as 2 Atari environments, that scaling networks sizes lead to decreasing performance because of gradient pathologies. They propose two interventions 1. multi-skip connections and 2. using the KFAC optimizer which is a second-order optimizer. Their interventions are evaluated across mutliple environments and algorithms successfully showing scalability.

**Questions:**

- Why did the authors specifically choose these two interventions? Did they also try other interventions to improve the gradient pathologies?
- Could the authors clarify which confidence intervals were used for the plots in the main part of the paper?

**Ethical Concerns:**

["NO or VERY MINOR ethics concerns only"]

**Final Justification:**

The authors provided a good rebuttal answering my open questions in great detail. I increased my score to accept.

**Limitations:**

yes

**Quality:**

3

**Strengths And Weaknesses:**

**Strengths**
- The paper has a clear motivation and is well written and easy to follow
- The authors provide extensive empirical evaluations both for the motivation as well as their interventions
- The proposed interventions are simple and easy to implement, making it easy for others to adapt

 **Weaknesses**
- The paper is purely empirical and does not provide a theoretical contribution
- As a nitpick, while overall the plots are very nice, the readability should be improved by properly scaling the text to the template. I find the text to be relatively small in many of the plots.
- I believe the authors are missing a few references in their related work:

CrossQ [1] showed that scaling the critic to 2048 neurons, in combination with their batch normalization, results in significant sample efficiency improvements. As such it should be cited among the other approaches for scaling parameters in deep RL.

And both [2] and [3] investigate scaling deep RL (mostly in terms of compute); however, similarly to Lyle et al. 2024, they identify growing parameter norms and gradient problems to be the bottleneck and introduce architectural interventions to stabilize the gradients. As such, these works should be discussed as well.

[1] Bhatt et al. CrossQ: Batch Normalization in Deep Reinforcement Learning for Greater Sample Efficiency and Simplicity. ICLR 2024

[2] Palenicek et al. Scaling Off-Policy Reinforcement Learning with Batch and Weight Normalization. 2025

[3] Lee et al. Hyperspherical Normalization for Scalable Deep Reinforcement Learning. ICML 2025

---

> ### Author Rebuttal · Authors · 2025-07-31
>
> We thank the reviewer for their feedback! We are happy that the reviewer found that “the paper is well written and easy to follow”, and that it “provides extensive empirical evaluations”. We are  also glad the reviewer recognized that the "interventions are simple and easy to implement, making it easy for others to adapt”. We respond to their main concerns below.
>
> # As a nitpick, while overall the plots are very nice, the readability should be improved by properly scaling the text to the template. I find the text to be relatively small in many of the plots.
>
> Thank you for pointing this out. We agree that readability is important and will ensure that all figures are properly scaled to meet the template guidelines in the final version.
>
> # I believe the authors are missing a few references in their related work ..
>
> Thank you for pointing us to these relevant works. We agree that they are highly relevant to the discussion on scaling in deep RL and will include them in the final version. While these works focus on batch/weight normalization and architectural constraints, our method offers a complementary and distinct approach by identifying multi-skip connectivity and Kronecker-factored optimization as synergistic mechanisms that improve both gradient propagation and parameter efficiency. In the revised manuscript, we will discuss these papers as approaches that analyze optimization pathologies in the context of scaling and propose architectural interventions to stabilize deep RL training.
>
> # Why did the authors specifically choose these two interventions? Did they also try other interventions to improve the gradient pathologies?
>
> Thank you for raising this important point. We selected Kronecker-factored optimization (Kron) and multi-skip connectivity based on the outcome of a systematic exploration of several candidate interventions aimed at improving gradient flow, plasticity, and stability in deep RL.
>
> As part of our investigation, we evaluated a wide range of alternative methods from prior work that aim to mitigate optimization pathologies in deep RL [2,10,11]. The methods were chosen because they have been used in other domains (e.g., supervised learning) and also deep RL to alleviate similar gradient-related issues. We excluded ReDo [1], Pruning [2], and Reset [3] as they require extensive hyperparameter tuning and, based on [2], their performance collapsed or stagnated when increasing network capacity. The evaluated methods include:
>
> - Second-order and adaptive optimizers: Apollo [4], Shampoo [5], AdaBelief [6]
>
> - Regularizers: L2 norm penalties [7], weight clipping [8], weight decay
>
> - Activation functions: GELU, CReLU [9]
>
> - Learning rate schedules: Cosine annealing and cyclic schedulers.
>
> - Learning rates at different scales: we try both multiplying and dividing the default learning rate (2.5e-4) by 12 to potentially compensate for the increased networks scale.
>
> As summarized in the table below (3 seeds, mean performance), none of these interventions consistently showed improved performance compared to our proposed combination.
>
> | Environment         | Ours    | Cosine LR Scheduler | Cyclic LR Scheduler | GELU activation | CReLU activation | L2 Norm  | Weight Clip | Weight Decay | lr=0.003 | lr=2.083e-05 |
> |---------------------|---------|---------------------|---------------------|-----------------|------------------|----------|-------------|--------------|---------------------|-------------------------|
> | Amidar‑v5           | **0.355897** | -0.001663          | 0.002276            | 0.015143        | 0.016514         | 0.029293 | 0.012283    | 0.013013     | 0.018848            | 0.078660                |
> | BattleZone‑v5       | **0.694566** | 0.031297           | -0.000287           | 0.015505        | -0.004594        | -0.004594| 0.035604    | 0.061446     | 0.019812            | 0.575407                |
> | Bowling‑v5          | **0.231831** | 0.050145           | -0.008721           | 0.028706        | -0.009811        | -0.000727| 0.030523    | 0.051235     | -0.086483           | 0.067587                |
> | DoubleDunk‑v5       | **-1.318182**| -2.318182          | -2.318182           | -2.363636       | -2.318182        | -2.363636| -2.454545   | -2.454545    | -2.454545           | -2.090909               |
> | Frostbite‑v5        | **0.881087** | -0.001921          | 0.027708            | 0.000890        | -0.010938        | 0.004755 | 0.075372    | 0.021736     | 0.008034            | 0.110036                |
> | KungFuMaster‑v5     | 0.975251 | -0.011055          | 0.003181            | -0.000823       | -0.011500        | -0.011278| -0.011500   | -0.011055    | -0.011055           | **1.362077**             |
> | NameThisGame‑v5     | **1.242066** | -0.030191          | -0.132767           | -0.261400       | -0.198430        | -0.068147| -0.197995   | -0.148227    | -0.134417           | 0.747251                |
> | Phoenix‑v5          | **0.655141** | 0.072687           | -0.087237           | -0.090014       | -0.090168        | -0.068722| -0.090940   | -0.093949    | -0.102358           | 0.541968                |
> | Qbert‑v5            | **1.094894** | 0.013814           | 0.010052            | -0.002833       | 0.008359         | 0.006102 | 0.016259    | 0.000177     | 0.001305            | 0.112282                |
> | Riverraid‑v5        | **0.993568** | -0.000127          | 0.345100            | 0.304636        | -0.056846        | -0.058494| 0.293514    | -0.044456    | -0.057765           | 0.237365                |
>
> Particularly for second order optimizers, one of our main interventions, we explored Apollo, Shampoo, and AdaBelief, as well as different architectural variants like ResNets and DenseNets. These results are included in the appendix (see Figure 13 for optimizer comparisons and Section D.3). Despite extensive hyperparameter tuning across both architectures and optimizer, these alternatives did not consistently improve trainability in deep RL.
>
> In contrast, our proposed combination of Kronecker-factored optimization (Kron) and multi-skip connections consistently outperformed these baselines. To strengthen this claim, we ran additional experiments for the rebuttal on the Atari-10 benchmark [12], comparing skip connectivity patterns when paired with Kron. Results show that richer skip connectivity is generally beneficial, with our multi-skip design performing best:
>
> | Method | ResNet + Kron | Multi-skip + Kron | DenseNet + Kron |
> |--------|---------------|--------------------|------------------|
> | **PPO** | 0.66          | **1.22**          | 0.69             |
> | **PQN** | 0.87          | **0.92**          | 0.90             |
>
> Table: Human-normalized scores on Atari-10 [12]. More skip connections generally improve performance, with multi-skip connections yielding the best results.
>
> We will add a brief discussion in the paper clarifying that these interventions were selected based on systematic experimentation, as documented in the appendix.
>
> # Could the authors clarify which confidence intervals were used for the plots in the main part of the paper?
>
> Thank you for the question. All plots in the main paper for deep RL experiments report 95% confidence intervals based on 3 independent seeds, following standard practice in recent deep RL literature [13]. We will clarify this in the figure captions and main text in the final version to avoid ambiguity.
>
> In addition to per-task learning curves, we report aggregate performance metrics across tasks, including interquartile mean (IQM), median [13]. This aggregate result provides a more reliable estimate of overall performance and demonstrates that our improvements are statistically significant and consistent across environments.
>
> ---
> *Thank you for the thorough review. Please let us know if any points remain unclear.*
>
> # References
>
> [1]. JSO Ceron et al, In value-based deep reinforcement learning, a pruned network is a good network , ICML’24
>
> [2]. G Sokar et al, The dormant neuron phenomenon in deep reinforcement learning, ICML’23
>
> [3]. E Nikishin et al, The primacy bias in deep reinforcement learning, ICML’22
>
> [4]. Xuezhe,  Apollo: An Adaptive Parameter-wise Diagonal Quasi-Newton Method for Nonconvex Stochastic Optimization, arXiv’21
>
> [5]. Vineet Gupta et al, Shampoo: Preconditioned Stochastic Tensor Optimization, arXiv’18
>
> [6]. Juntang Zhuang et al, Adabelief optimizer: Adapting stepsizes by the belief in observed gradients, NeurIPS’20
>
> [7]. A Kumar et al, Offline Q-Learning on Diverse Multi-Task Data Both Scales And Generalizes, ICLR’23
>
> [8]. Mohamed Elsaye et al, Weight clipping for deep continual and reinforcement learning, RLC’24
>
> [9]. Zaheer et al, Loss of Plasticity in Continual Deep Reinforcement Learning, CoLLAs’23
>
> [10]. Skander et al, No Representation, No Trust: Connecting Representation, Collapse, and Trust Issues in PPO, NeurIPS’24
>
> [11]. Arthur et al, A Study of Plasticity Loss in On-Policy Deep Reinforcement Learnin, NeurIPS’24
>
> [12]. Aitchison, Matthew, Penny Sweetser, and Marcus Hutter.  Atari-5: Distilling the arcade learning environment down to five games. ICML’23.
>
> [13]. R Agarwal, M Schwarzer, PS Castro, A Courville, MG Bellemare, Deep Reinforcement Learning at the Edge of the Statistical Precipice, NeurIPS’21

---

> > ### Comment · Reviewer_Uxog · 2025-08-01
> >
> > Thank you for your extensive rebuttal. I have raised my score to accept.

---

### Official Review · Reviewer_vtbr · 2025-07-03

**Clarity:** 3
**Significance:** 2
**Originality:** 3
**Rating:** 5
**Confidence:** 3

**Summary:**

In this work, the authors study the impact of scaling model parameters on the ability of models to learn and adapt to the nonstationary training rewards typically encountered during reinforcement learning. As the number of model parameters increases and the depth of the network increases, several well-known stability issues specifically hinder RL networks' ability to learn effectively. The address these issues the authors propose and study the impact of adding more long-range connections to propagate gradient information over long depths and incorporating higher-order optimizers to utilize better informed gradient updates during training. Empirically, the authors show that in the ALE learning environment that these updates result in improved gradient stability for large networks and generate networks is higher rewards in several ALE environments.

**Questions:**

How does the change in the skip length impact the training dynamics?

**Ethical Concerns:**

["NO or VERY MINOR ethics concerns only"]

**Final Justification:**

Based on the author's thorough rebuttal and comments regarding the input from other reviewers, I will raise my score.

**Limitations:**

The limitations of the proposed method were outlined by the authors adequately in section 7 of the paper. Specifically, the study is conducted over a relatively small range of network sizes.

**Paper Formatting Concerns:**

I did not observe any formatting issues.

**Quality:**

3

**Strengths And Weaknesses:**

Strengths:
--------------
1. This work is well-written with figures that illustrate visually the author's intentions and complement the writing seamlessly. I particularly enjoyed the figures in Section 4 that displayed the mean gradient norms and normalized scores.
2. The topic of scaling RL to larger sizes is an interesting line of questioning, and the authors offer a systematic study of the impact of the 2 changes on the ability of these large networks to propagate gradient information effectively.
3. Results are provided on several environments from the ALE environment and Issac Gym for a number of network architectures and using different training algorithms.
4. Although the techniques are not novel, the application of them to stabilize the gradient flow during RL training appears to be a great application of which I was not previously aware. Overall, this study appears to be an incremental improvement on existing training strategies and provides interesting examples to improve the performance of RL models for networks and a large number of parameters.

Weaknesses:
------------------
1. Neither of the proposed methods to stabilize the performance is novel and has indeed been studied extensively in several other contexts where the training dynamics were stable.
2. Of the 2 proposals, I believe usage of higher-order optimizers to be the least novel innovation. I would assume in most contexts, integrating higher-order information would improve training and would be the default if it were not considerably more computationally expensive. This aspect is exacerbated by the fact that the authors are interested in scaling up the model size, which would make the application of higher-order optimization methods even more precarious.
3. Skip connections were studied extensively in the context of CNNs, and it's not clear from the study what the minimum the skip connections should be to ensure gradient information is passed effectively.

---

> ### Author Rebuttal · Authors · 2025-07-31
>
> We sincerely appreciate the reviewer’s valuable comments. We are happy that the reviewer found that “the paper is well-written and clear”. We're also pleased that the reviewer recognized “the systematic nature of our study” and that we present results “on several environments from the ALE environment and Issac Gym for a number of network architectures and different RL algorithms”. We respond to their main concerns below.
>
> # Neither of the proposed methods to stabilize the performance is novel and has indeed been studied extensively in several other contexts where the training dynamics were stable.
>
> We agree that the individual techniques we employ, such as architectural modifications and second-order optimization, have been explored in other domains with more stable training dynamics. Our contribution lies in demonstrating how these techniques, when adapted thoughtfully, can address fundamental challenges in deep RL: the implicit non-stationarity that undermines plasticity and learning stability; a problem that is further exacerbated as model scale increases [1,2,3]. We approach this through targeted gradient interventions, which we show to be highly effective across diverse tasks and rl algorithms.
>
> While some components are well-known individually, our multi-skip architecture, which broadcasts CNN features to all downstream layers (see Figure 4), proved particularly effective for improving gradient flow under non-stationarity conditions. We believe the thoroughness of our experimentation and analysis makes these findings a valuable foundation for future research in deep RL at scale.
>
> # Of the 2 proposals, I believe usage of higher-order optimizers to be the least novel innovation. I would assume in most contexts, integrating higher-order information would improve training  ...
>
> We agree that incorporating higher-order information should improve training in principle, and that computational cost remains a key barrier. However, in deep RL, simply introducing second-order methods does not guarantee improvements. In fact, our experiments with other second-order optimizers such as Shampoo [4] and Apollo [5], despite extensive hyperparameter tuning, did not yield gains (see Figure 13 in the appendix). This suggests that applying second-order optimization in deep RL is non-trivial.
>
> Our analyses and systematic experimentation were key in identifying Kronecker-factored optimization as particularly effective for improving trainability under non-stationary settings. We believe this careful empirical investigation offers valuable insights for future work aiming to move beyond the standard reliance on Adam in deep RL and towards scaling deep RL.
>
> # Skip connections were studied extensively in the context of CNNs, and it's not clear from the study what the minimum the skip connections should be to ensure gradient information is passed effectively. How does the change in the skip length impact the training dynamics?
>
> We appreciate the reviewer’s insightful comment regarding the importance of understanding the minimum number and placement of skip connections required to ensure effective gradient propagation. While our original submission did not include a detailed ablation on this point, we have now conducted additional experiments to specifically probe this question.
>
> To isolate the effect of skip length on performance, we fix the main network of our proposed MultiSkip architecture (in *large* size, which includes 5 residual blocks) and vary how many of these blocks receive skip connections from the encoder. When Skip = k, we apply the encoder features as skip connections to the first k residual blocks immediately following the encoder, while the remaining (5 - k) blocks operate without direct encoder input. The table below reports human-normalized scores on the Atari-10 benchmark [6]:
>
> | Environment         | Skip=1 | Skip=2 | Skip=3 | Skip=4 | Skip=5 |
> |---------------------|--------|--------|--------|--------|--------|
> | Amidar‑v5           | 0.20   | 0.17   | 0.19   | 0.20   | **0.36** |
> | BattleZone‑v5       | 0.01   | 0.67   | 0.62   | 0.60   | **0.69** |
> | Bowling‑v5          | 0.07   | 0.04   | 0.04   | 0.08   | **0.23** |
> | DoubleDunk‑v5       | -2.09  | -2.00  | -1.55  | -1.36  | **-1.32** |
> | Frostbite‑v5        | 0.67   | 0.70   | 0.79   | **0.92** | 0.88   |
> | KungFuMaster‑v5     | 0.93   | 0.95   | 0.93   | **1.12** | 0.98   |
> | NameThisGame‑v5     | 0.79   | 0.65   | 0.72   | 0.85   | **1.24** |
> | Phoenix‑v5          | **0.69** | 0.68   | 0.68   | 0.66   | 0.66   |
> | Qbert‑v5            | 0.84   | 1.01   | **1.06** | 0.91   | 1.09   |
> | Riverraid‑v5        | 0.42   | 0.44   | 0.65   | 0.69   | **0.99** |
> | **Aggregate (mean)**| 0.25   | 0.33   | 0.41   | 0.47   | **0.58** |
>
> Performance steadily improves as more skip connections are added, peaking when all 5 blocks are connected. This supports our original design decision to broadcast features to all MultiSkip blocks.
>
> Furthermore, our main paper already demonstrates consistent gains when skip connections are present in the CNN network. In particular, Figure 9 reports large improvements when using the Impala CNN architecture (which includes residual connections) compared to a plain CNN.
>
> ---
> *Thank you for the thoughtful review. We hope these additions address the main concerns regarding novelty and applicability. Please let us know if any points remain unclear.*
>
> # References
>
> [1]. Guozheng Ma et at, Network Sparsity Unlocks the Scaling Potential of Deep Reinforcement Learning, ICML’25
>
> [2]. JSO Ceron et al, In value-based deep reinforcement learning, a pruned network is a good network, ICML’24
>
> [3]. JSO Ceron et al, Mixtures of Experts Unlock Parameter Scaling for Deep RL, ICML’24
>
> [4]. Vineet Gupta et al, Shampoo: Preconditioned Stochastic Tensor Optimization, arXiv’18
>
> [5]. Xuezhe,  Apollo: An Adaptive Parameter-wise Diagonal Quasi-Newton Method for Nonconvex Stochastic Optimization, arXiv’21
>
> [6]. Aitchison, Matthew, Penny Sweetser, and Marcus Hutter. Atari-5: Distilling the arcade learning environment down to five games. ICML’23.

---

> > ### Comment · Reviewer_vtbr · 2025-08-05
> >
> > Thank you for the thorough response to my inquiries and to the other reviewers. I have raised my score accordingly.

---

### Official Review · Reviewer_rJWh · 2025-07-03

**Clarity:** 4
**Significance:** 3
**Originality:** 2
**Rating:** 5
**Confidence:** 4

**Summary:**

This work considers how to make Deep RL scale better with network size. They highlight that RL objectives are non-stationary due to both a drift in the TD-error target as well as a changing policy. Then, due to poor plasticity effects of some neural networks, learning can get stalled. They examine this effect also on supervised learning tasks that also show this effect. They claim that the issue is partially due to vanishing gradient magnitudes, and show that gradient magnitudes becomes smaller when increasing network depth or width, and learning performance also degrades. To tackle these problems they use skip connections and also a Kronecker factored optimizer. These techniques remedy the issue and greatly improve performance in both RL tasks and the supervised learning task that they considered. Experimentally, they were mainly working with the PQN algorithm (a recent scalable and strong parallelized Q-learning based algorithm), and their interventions lead to around 80% improvement in performance. They also combined their method with the IMPALA CNN architecture, and showed an improvement there as well, moreover, they considered combining the Kron optimizer with the SIMBA architecture (that already has skip connections) and SIMBA also achieved better scaling with the network size and significantly improved performance. Moreover, combining their methods with PPO also lead to better scaling and improved performance.

**Questions:**

In Figure 5 (right), is the gradient magnitude for +Kron the magnitude of the update vector (i.e., inverse curvature * gradient), or is it just the magnitude of the gradient? To clarify, does the optimizer lead to different regions in the optimization space where the gradient does not become small, or does it just add a preconditioner to the gradient that adjusts the magnitude?

“significantly boots performance” → significantly boosts performance

**Ethical Concerns:**

["NO or VERY MINOR ethics concerns only"]

**Final Justification:**

The authors responded to my questions, and I believe that review remains adequate, and the paper is a clear accept. The authors argued about the novelty, but I believe my view regarding the novelty remains adequate as the novelty appears marginal without any groundbreaking ideas. The response regarding the gradient magnitudes was interesting.

**Limitations:**

Yes.

**Paper Formatting Concerns:**

No.

**Quality:**

3

**Strengths And Weaknesses:**

Strengths

* They show strong performance improvements.

* The paper is clearly written.

* The results are consistent across many different architecture types and algorithms.

Weaknesses

* The novelty is not that high as skip connections are well-known, the Kron optimizer is an existing one, and gradient/plasticity issues are discussed in other works as well.

* Using the Kron optimizer introduces a significant computational overhead (e.g., for the large network size setting, the computation time goes from 1h27min to 7h 42min in Table 1). They have discussed this in their limitation section.

* The gradient magnitudes becoming smaller may be coincidental to the performance dropping; it did not seem proven that performance drops because the gradient magnitudes are small.


Overall, this is a strong well-written paper that shows how to improve performance via better scalability in Deep RL.

---

> ### Author Rebuttal · Authors · 2025-07-31
>
> We thank the reviewer for their feedback! We are happy that the reviewer found that “the paper is clearly written” and that “the results are consistent across many different architecture types and algorithms”. We respond to their main concerns below.
>
> # The novelty is not that high as skip connections are well-known, the Kron optimizer is an existing one, and gradient/plasticity issues are discussed in other works as well.
>
> While the individual components (skip connections, Kronecker-factored optimization, and plasticity analysis) are based on existing ideas, our contributions lie in how we integrate and apply them in a deep RL context to address specific failure modes, particularly those involving gradient degradation.
>
> While Kronecker-factored methods (e.g., [1]) have been explored before, they are not commonly adopted in deep RL. Our contribution includes a careful and extensive empirical study of using a Kronecker-factored optimizer within modern deep RL. We position this as a step toward broadening the toolkit for deep RL practitioners.
>
> For architectures, we extend prior work on skip connections with our multi-skip network, which broadcasts CNN-extracted features to all subsequent layers. To our knowledge, this form of broadcasting has not been explored in deep RL, making the design novel. Figure 4 illustrates its structure relative to standard ResNet-style MLPs and DenseNets. Importantly, these multi-skip connections emerged as an effective mechanism for mitigating the vanishing gradient pathologies identified in our analysis. We have updated the paper to make the novelty clear.
>
> # The gradient magnitudes becoming smaller may be coincidental to the performance dropping; it did not seem proven that performance drops because the gradient magnitudes are small.
>
> While asserting a direct causal relationship is difficult, we do observe a strong correlation between vanishing gradients and performance drop. The fact that performance increases when we mitigate vanishing gradients strengthens this connection. Nonetheless, we will clarify in the final version that this correlation does not explicitly imply causality.
>
> # In Figure 5 (right), is the gradient magnitude for +Kron the magnitude of the update vector (i.e., inverse curvature * gradient), or is it just the magnitude of the gradient?
>
> It represents the magnitude of the preconditioned gradient. That is, the update vector resulting from the multiplication of the gradient by the preconditioning matrix.
>
> # To clarify, does the optimizer lead to different regions in the optimization space where the gradient does not become small, or does it just add a preconditioner to the gradient that adjusts the magnitude?
>
> This is a great question, but quite difficult to answer assertively, particularly given the high levels of non-stationarity in deep RL (which result in shifting optimization landscapes). We hypothesize that the improvements we observe are primarily due to better pre-conditioning, though a deeper understanding remains an open question. We agree this is an important direction for future investigation and we will add a note in the discussion accordingly.
>
> ---
> *Thank you for the thorough review. Please let us know if any points remain unclear.*
>
> # References
>
> [1]. Firouzi, Mohammad. KF-LAX: Kronecker-factored curvature estimation for control variate optimization in reinforcement learning. arXiv’18.

---

> > ### Comment · Reviewer_rJWh · 2025-08-01
> >
> > Thank you for the response. I have no further questions.

---

### Official Review · Reviewer_5mBc · 2025-07-18

**Clarity:** 4
**Significance:** 4
**Originality:** 4
**Rating:** 6
**Confidence:** 3

**Summary:**

This paper looks to address a major challenge in reinforcement learning. Namely, how to scale deep RL algorithms to larger architectures and datasets. As shown in prior work, RL does not scale in the same respects that generative modeling nor supervised learning have scaled. There are many theories as to why and yet this problem remains poorly understood. The authors highlight the core challenge is the inherent non-stationarity of the training objective in RL.

The authors proposed looking at how gradients propagate throughout the network during training, rather than modifying a specific RL algorithm or aspects of RL training. The authors note that this approach is well-studied in supervised learning, but unexplored in RL. (I am unaware of anyone addressing this issue in RL, but my expertise is also outside RL. It would surprise me if no one studied this before.)

The contribution of the paper is an extensive study of how gradient decompositions interact with non-stationary problems. Their proposed approach is to actively control gradient propagation in the network, through both second-order optimization methods and multi-skip residual connections. They show that this approach to stable gradients can greatly improve a variety of RL algorithms across a variety of environments. This is a promising line of research that could unlock the potential of scaling RL algorithms to larger architectures.

Overall, this paper seems thorough and well-presented. It introduces a highly relevant problem, namely that of understanding why RL algorithms fail when networks are scaled up. It provides excellent evidence for this problem in (Section 3), reasonable and understandable solutions (Section 4), and comprehensive experiments (Section 4-5, Appendix).

**Questions:**

- Is it true that only MLP’s (aside from convnet image encoders for ALE) were studied in this paper? How would you apply these approaches to different architectures (transformers, diffusion policy, etc)? Or is more research needed?

- Can the approach be augmented for different rates of change in stationarity? How much non-stationarity can the new approach tolerate before failure?

**Ethical Concerns:**

["NO or VERY MINOR ethics concerns only"]

**Final Justification:**

I think this is a great paper taking insights from the generative modeling and supervised learning community to tackle an essential problem for getting RL to scale to larger networks. While I'm not from the RL field, I've known many groups trying to address this difficult problem themselves and have commented on the need to having something that scales to larger networks in RL. Other groups have shared a dissatisfaction with current approaches of getting RL to scale to larger networks, which makes me excited to see a work scale RL to large networks so robustly. The experiments in the paper were thorough and well-motivated and improved during the discussion period. Other reviewers brought up concerns for the degree of novelty given many of the ideas being discovered in these other fields. I personally do not think that's too important in this context. The paper does a great job building off ideas from disparate fields and getting them to work in more challenging non-stationary domains. Reading the paper gives me confidence the ideas work and generalize.

**Limitations:**

yes

**Paper Formatting Concerns:**

Line 91: dificulty -> difficulty
Figure 1: The lightly shaded boxes were a bit confusing as they look like they are being hidden due to their lighter shade. Maybe use solid black lines and refer to them as the lightly-colored boxes?
Figure 2: What MLP width is used in the two plots on the left?
Line 219: boots -> boosts

**Quality:**

4

**Strengths And Weaknesses:**

- The paper is well-written and clear.
- The paper could use a front figure on the second page to highlight the key mechanisms of the approach.
- The authors give a good review of deep reinforcement learning as well as gradient propagation and stability.
- The paper is friendly to readers from both a supervised learning and reinforcement learning background.
- The experiments to study this problem are extensive. Ranging from the full ALE and several RL algorithms.
- The improvements due to gradient stabilization are large on several of the environments. Though I’m someone surprised that the approach doesn’t lead to reduced variance in outcome.
- As the authors note, the study is computationally expensive. Due to this, the networks sizes were not scaled beyond 120 million parameters. Given the popularity of RLHF on large language models, it would be useful to see a single run of the approach on a network of similar scale to see if performance improves.
- A question not addressed in this paper is the rate of change in stationarity. Small and large changes in stationarity may affect the underlying training algorithm differently. On different RL problems or algorithms, the data collected from the policy can change at different rates even during the same run of RL training. It’s not clear to me how this approach could detect and address this aspect of the problem.

---

> ### Author Rebuttal · Authors · 2025-07-31
>
> We thank the reviewer for their feedback! We are happy that the reviewer found that “the paper is well-written and clear” and that “the experiments to study this problem are extensive”. We respond to their main concerns below.
>
> # **Q1) Is it true that only MLP’s (aside from convnet image encoders for ALE) were studied in this paper?**
>
> Yes. Our experiments focus on combinations of convolutional backbones (e.g. to process ALE frames) and fully‑connected heads i.e., MLP architectures, mirroring the vast majority of prior model‑free deep RL work [1,2,3].
>
> # **Q2) How would you apply these approaches to different architectures (transformers, diffusion policy, etc)? Or is more research needed?**
>
> Kron and standard normalization layers extend directly to any layer whose parameters or activations admit a preconditioning or normalization step: transformers, LSTMs, diffusion‑policy networks, and others.
>
> Our multi‑skip broadcast mechanism relies on propagating intermediate feature maps forward to all downstream layers. While the concept could extend to attention-based models (e.g. broadcasting early Transformer block outputs to later ones), deciding what to broadcast (keys, values, token embeddings, etc.) and how to merge them meaningfully (residual, cross-attention, etc.) is non-trivial. We expect this to require additional research to identify the right injection and aggregation strategies in attention‑based or recurrent settings. We view this as promising future work, and we will add a discussion around this in the final version.
>
> We also ran PQN with LSTM policies on the full ALE benchmark (57 games, 200M frames) to evaluate whether our gradient interventions generalize to recurrent architectures. The table below shows improvements of Kron + **MultiSkip** + LSTM  over Adam + **MLP** + LSTM across all environments.
>
> | **Statistic**                     | **Value**    |
> |-----------------------------------|-------------:|
> | Average Improvement (%)           | 1558.23      |
> | Median Improvement (%)            | 86.93        |
> | Games with Improvement            | 48 / 55      |
> | Percentage of Games Improved (%)  | 87.27        |
> | Maximum Improvement (%)           | 43,844.44    |
> | Minimum Improvement (%)           | -219.38      |
>
> Due to space constraints, we cannot include the full per-game results in the main text or in the rebuttal, but we will add them to the appendix, along with:
>
> - Per-game learning curves (similar to Figs. 16 and 17)
>
> - Per-game % improvements (similar to Fig. 6, right).
>
> # **Q3) Can the approach be augmented for different rates of change in stationarity? How much non-stationarity can the new approach tolerate before failure?**
>
> Yes, our approach generalizes across a range of non-stationarity regimes, and our results suggest it performs robustly under varying degrees of stationarity.
>
> Previously, we demonstrated this in online RL experiments, where non-stationarity arises primarily from  policy-induced distribution shifts and target bootstrapping. To further evaluate the generality of our method across different degrees of non-stationarity, we introduced new **offline RL experiments** where the only form of non-stationarity comes from bootstrap targets (i.e., no policy-induced drift, but still changes in the network targets for actions after Bellman backups propagate), using a challenging benchmark from a recent study [4]. Park, Seohong, et al. (2025) showed that standard offline Q-learning baselines fail even with 100M samples and near-perfect coverage.
>
> We retrained those same baselines with and without our Kron + Multi-Skip interventions. Across nearly all environments and algorithms, our method consistently improves performance, even in these highly brittle offline settings (see table below). Notably, in difficult environments like humanoidmaze-giant-navigate, gains were substantial (in many cases even over 90%).
>
> | **Algorithm**                 | **cube-octuple-play** | **humanoidmaze-giant-navigate** | **puzzle-4x5-play**  | **puzzle-4x6-play**  |
> |-------------------------------|-----------------------|---------------------------------|----------------------|----------------------|
> | **contrastive-rl (Baseline)** | 0.000 ± 0.000         | -                               | 0.060 ± 0.085        | 0.047 ± 0.066        |
> | **contrastive-rl (Ours)**     | 0.000 ± 0.000         | -                               | 0.047 ± 0.047        | 0.013 ± 0.019        |
> | ---                           |                       |                                 |                      |                      |
> | **dsharsa (Baseline)**        | **0.260 ± 0.028**     | 0.713 ± 0.009                   | 0.973 ± 0.019        | 0.727 ± 0.028        |
> | **dsharsa (Ours)**            | 0.100 ± 0.009         | **0.953 ± 0.028**               | **0.993 ± 0.009**    | **0.853 ± 0.019**    |
> | ---                           |                       |                                 |                      |                      |
> | **gc-fbc (Baseline)**         | 0.000 ± 0.000         | 0.040 ± 0.019                   | **0.020 ± 0.009**    | 0.007 ± 0.009        |
> | **gc-fbc (Ours)**             | 0.000 ± 0.000         | **0.267 ± 0.038**               | 0.000 ± 0.000        | 0.000 ± 0.000        |
> | ---                           |                       |                                 |                      |                      |
> | **gc-fql (Baseline)**         | 0.000 ± 0.000         | 0.053 ± 0.000                   | 0.160 ± 0.057        | 0.187 ± 0.019        |
> | **gc-fql (Ours)**             | 0.000 ± 0.000         | **0.920 ± 0.057**               | **0.600 ± 0.566**        | **0.647 ± 0.405**    |
> | ---                           |                       |                                 |                      |                      |
> | **gc-iql (Baseline)**         | 0.000 ± 0.000         | -                               | 0.200 ± 0.000        | 0.100 ± 0.047        |
> | **gc-iql (Ours)**             | 0.000 ± 0.000         | -                               | 0.200 ± 0.000        | **0.213 ± 0.094**    |
> | ---                           |                       |                                 |                      |                      |
> | **gc-sac+bc (Baseline)**      | 0.000 ± 0.000         | 0.347 ± 0.207                   | 0.187 ± 0.019        | 0.100 ± 0.047        |
> | **gc-sac+bc (Ours)**          | 0.000 ± 0.000         | **0.880 ± 0.019**               | 0.193 ± 0.009        | **0.333 ± 0.000**    |
> | ---                           |                       |                                 |                      |                      |
> | **h-fbc (Baseline)**          | 0.253 ± 0.057         | 0.260 ± 0.009                   | 0.053 ± 0.019        | 0.047 ± 0.028        |
> | **h-fbc (Ours)**              | 0.280 ± 0.038         | **0.360 ± 0.000**               | **0.173 ± 0.019**    | **0.180 ± 0.104**    |
> | ---                           |                       |                                 |                      |                      |
> | **h-iql (Baseline)**          | 0.067 ± 0.000         | 0.673 ± 0.009                   | 0.093 ± 0.019        | 0.120 ± 0.000        |
> | **h-iql (Ours)**              | **0.367 ± 0.009**     | **0.807 ± 0.047**               | **0.227 ± 0.019**    | 0.133 ± 0.038        |
> | ---                           |                       |                                 |                      |                      |
> | **n-gc-sac+bc (Baseline)**    | 0.000 ± 0.000         | 0.847 ± 0.009                   | **0.907 ± 0.000**    | 0.300 ± 0.066        |
> | **n-gc-sac+bc (Ours)**        | 0.000 ± 0.000         | **0.920 ± 0.019**               | 0.547 ± 0.038        | **0.780 ± 0.028**    |
> | ---                           |                       |                                 |                      |                      |
> | **sharsa (Baseline)**         | **0.427 ± 0.038**     | 0.533 ± 0.057                   | 0.893 ± 0.075        | 0.653 ± 0.075        |
> | **sharsa (Ours)**             | 0.280 ± 0.019         | **0.813 ± 0.038**               | **1.000 ± 0.000**    | **0.887 ± 0.047**    |
>
> *Baseline algorithms are implemented in [4]. Ours corresponds to adding Kron + MultiSkip networks. gc- means “goal-conditioned” and h- means “hierarchical”*
>
> Combined with the online RL results in the main paper, which involve compounded non-stationarity from both policy and value updates, these new results show that our approach:
> Is effective across different rates and sources of non-stationarity.
> Delays or prevents failure even under conditions where standard methods collapse.
> Our method not only tolerates significant non-stationarity, but also improves stability under a broad spectrum of dynamics.
>
>
> # **Paper Formatting Concerns**
>
> *Thanks a lot for bringing these points to our attention. We will incorporate all suggested corrections into the final version of the paper.*
>
>
> # **References**
>
> [1]. Obando-Ceron, J., Sokar, G., Willi, T., Lyle, C., Farebrother, J., Foerster, J. N., Dziugaite, G., Precup, D., and Castro, P. S.. Mixtures of experts unlock parameter scaling for deep rl. ICML’24
>
> [2]. Lyle et al. "Understanding plasticity in neural networks." ICML’23
>
> [3]. V Mnih et al. Human-level control through deep reinforcement learning, Nature’ 15
>
> [4]. Park, Seohong, et al. Horizon Reduction Makes RL Scalable, arXiv’25.

---

> > ### Comment · Reviewer_5mBc · 2025-08-04
> >
> > Thank you for the thorough response and addressing my concerns. I appreciate the addition of offline RL experiments to address the degrees of non-stationarity problem. In my eyes, this remains a great paper that was strengthened by the rebuttal.

---

### Author Response · Authors · 2025-08-09
**General Response**

We greatly value the reviewers’ thoughtful feedback and active engagement during the rebuttal phase. The detailed comments and constructive suggestions have not only helped us clarify our contributions and address potential weaknesses, but have also strengthened the paper’s overall impact. We are grateful for the time and effort invested in this process, and we are excited about the improved clarity and significance of our work.

Thanks!
- Authors

---

### Decision · Program_Chairs · 2025-09-17

**Decision:**

Accept (spotlight)

**Comment:**

This paper aims to tackle the challenges involved with scaling RL algorithms. Reviewers uniformly praised the writing, presentation, and thoroughness of the paper, and were all enthusiastic about seeing it at the conference.